# Beyond ImageNet Attack: Towards Crafting Adversarial Examples for Black-box Domains

**Qilong Zhang[1]\*, Xiaodan Li[2], Yuefeng Chen[2], Jingkuan Song[1]†,**
**Lianli Gao[1], Yuan He[2], and Hui Xue[2]**
[1]University of Electronic Science and Technology of China, China
qilong.zhang@std.uestc.edu.cn, jingkuan.song@gmail.com, lianli.gao@uestc.edu.cn
[2]Alibaba Group, China
{fiona.lxd,yuefeng.chenyf,heyuan.hy,hui.xueh}@alibaba-inc.com

## ABSTRACT

Adversarial examples have posed a severe threat to deep neural networks due to their transferable nature. Currently, various works have paid great efforts to enhance the cross-model transferability, which mostly assume the substitute model is trained in the same domain as the target model. However, in reality, the relevant information of the deployed model is unlikely to leak. Hence, it is vital to build a more practical black-box threat model to overcome this limitation and evaluate the vulnerability of deployed models. In this paper, with only the knowledge of the ImageNet domain, we propose a Beyond ImageNet Attack (BIA) to investigate the transferability towards black-box domains (unknown classification tasks). Specifically, we leverage a generative model to learn the adversarial function for disrupting low-level features of input images. Based on this framework, we further propose two variants to narrow the gap between the source and target domains from the data and model perspectives, respectively. Extensive experiments on coarse-grained and fine-grained domains demonstrate the effectiveness of our proposed methods. Notably, our methods outperform state-of-the-art approaches by up to 7.71% (towards coarse-grained domains) and 25.91% (towards fine-grained domains) on average. Our code is available at `https://github.com/Alibaba-AAIG/Beyond-ImageNet-Attack`.

## 1 Introduction

Deep neural networks (DNNs) have achieved remarkable success in the image classification task in recent years. Nonetheless, advances in the field of adversarial machine learning (Szegedy et al., 2014; Goodfellow et al., 2015; Zhang et al., 2022) make DNNs no longer reliable. By adding a well-designed perturbation on a benign image (*a.k.a* adversarial attack), the resulting adversarial examples can easily fool state-of-the-art DNNs. To make the matter worse, the adversarial attack technique can even be applied in the physical world (Sharif et al., 2016; Kurakin et al., 2017a; Xu et al., 2020; Duan et al., 2021), which inevitably raises concerns about the stability of deployed models. Therefore, exposing as many "blind spots" of DNNs as possible is a top priority.

Generally, deployed models are mainly challenged with two threat models: white-box and black-box. For white-box threat model (Kurakin et al., 2017b; Moosavi-Dezfooli et al., 2016; Carlini & Wagner, 2017; Shi et al., 2019; Liu et al., 2022), the attacker can obtain complete knowledge of the target model, such as the gradient for any input. However, deployed models are usually opaque to unauthorized users. In this scenario, prior black-box works (Poursaeed et al., 2018; Dong et al., 2018; Xie et al., 2019; Inkawhich et al., 2020; Gao et al., 2020b; Wang et al., 2021) mostly assume that the source data for training the target model is available and mainly explore the cross-model transferability among models trained in the same data distribution. Specifically, perturbations are crafted via accessible white-box model (*a.k.a* substitute model), and resulting adversarial examples sometimes can fool other black-box models as well. Yet, these works still ignore a pivotal issue: *A*

---

\*Work done when Qilong Zhang interns at Alibaba Group, China
†Corresponding author

Table 1: A comparison of datasets from different domains.

| Dataset | Resolution | Type | Test size | Classes |
|---|---|---|---|---|
| ImageNet (Russakovsky et al., 2015) | 224×224 | - | 50,000 | 1,000 |
| CIFAR-10 (Krizhevsky, 2009) | 32×32 | coarse-grained | 10,000 | 10 |
| CIFAR-100 (Krizhevsky, 2009) | 32×32 | coarse-grained | 10,000 | 100 |
| STL-10 (Coates et al., 2011) | 96×96 | coarse-grained | 8,000 | 10 |
| SVHN (Netzer et al., 2011) | 32×32 | coarse-grained | 26,032 | 10 |
| CUB-200-2011 (Wah et al., 2011) | 448×448 | fine-grained | 5,740 | 200 |
| Stanford Cars (Krause et al., 2013) | 448×448 | fine-grained | 8,041 | 196 |
| FGVC Aircraft (Maji et al., 2013) | 448×448 | fine-grained | 3,333 | 100 |

*model owner is unlikely to leak the relevant information of the deployed model.* To overcome this limitation, query-based black-box attacks (Papernot et al., 2016; Brendel et al., 2018; Chen et al., 2020; Li et al., 2021) are proposed, which adjust adversarial examples just according to the output of the target model. However, the resource-intensive query budget is extremely costly and inevitably alerts the model owner.

Therefore, we need a more "practical" black-box threat model to address this concern, *i.e.*, without any clue about the training data distribution as well as the pre-trained model based on it, and even querying is forbidden. Intuitively, this threat model is more challenging to build for attackers and more threatening to model owners. To the best of our knowledge, a recent work called CDA (Naseer et al., 2019) is the first to attempt such an attack. Specifically, it learns a transferable adversarial function via a generator network against a different domain (training data and pre-trained model are all from ChestX-ray (Wang et al., 2017) domain). During inference, it directly crafts adversarial examples for benign ImageNet images to fool target ImageNet pre-trained models. However, its cross-domain transfer strength is still moderate. Besides, relying on small-scale datasets to train a generator may limit the generalization of the threat model.

Considering that ImageNet is a large-scale dataset containing most of common categories in real life and there are various off-the-shelf pre-trained models, one can easily dig out much useful information to build a strong threat model. Therefore, in this paper, solely relying on the knowledge of the ImageNet domain, we introduce an effective **Beyond ImageNet Attack (BIA)** framework to enhance the cross-domain transferability of adversarial examples. To reflect the applicability of our approach, we consider eight different image classification tasks (listed in Table 1). Figure 1 illustrates an overview of our method. Particularly, we learn a flexible generator network $\mathcal{G}_\theta$ against ImageNet domain. Instead of optimizing the domain-specific loss function like CDA, our method focuses on disrupting low-level features following previous literature to ensure the good transferability of our BIA. Furthermore, we propose two variants based on the vanilla BIA to narrow the gap between source and target domains. Specifically, from the data perspective, we propose a random normalization ($\mathcal{RN}$) module to simulate different data distributions; from the model perspective, we propose a domain-agnostic attention ($\mathcal{DA}$) module to capture essential features for perturbing. In the inference phase, our $\mathcal{G}_\theta$ accepts images of any domain as the input and crafts adversarial examples with one forward propagation. Extensive experiments demonstrate the effectiveness of our proposed methods. Towards the coarse-grained and fine-grained domains, we can outperform state-of-the-art approaches by up to 7.71% and 25.91% on average, respectively. Besides, our methods can also enhance the cross-model transferability in the source domain.

## 2 RELATED WORKS

**Iterative Optimization Approaches.** Under the black-box threat model, iterative attack methods are a popular branch, which usually adopt domain-specific loss or intermediate feature loss to craft adversarial examples. For the former, Madry et al. (2018) extend Goodfellow et al. (2015) to perform projected gradient descent from randomly chosen starting points inside $\epsilon$-ball. Dong et al. (2018) introduce momentum term to stable the update direction. Xie et al. (2019) apply random transformations of the input at each iteration, thus mitigating overfitting. Gao et al. (2020a) propose patch-wise perturbation to better cover the discriminative region. Wu et al. (2020a) explore the security weakness of skip connections (He et al., 2016; Huang et al., 2017) to boost attacks.

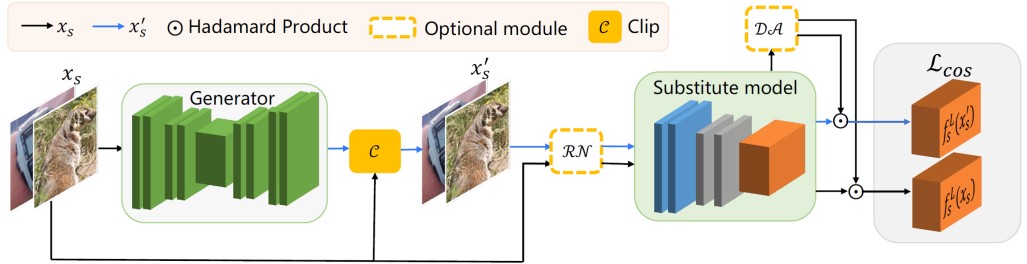

Figure 1: Our proposed generator framework aims to decrease the cosine similarity of feature between benign image $x_s$ and adversarial example $x_s'$ during the training phase. Training data and substitute model are all from the ImageNet domain. $\mathcal{C}$ module is applied to constrain $x_s'$ in the $\ell_\infty$-ball of $x_s$. $\mathcal{RN}$ and $\mathcal{DA}$ are optional, which can further improve the transferability.

Different from the methods mentioned above, intermediate feature-based methods focus on disrupting low-level features. For example, Zhou et al. (2018) maximize the Euclidean distance between the source image and target image in feature space and introduce regularization on perturbations to reduce variations. Inkawhich et al. (2019) make the source image close to the target image in feature space. Lu et al. (2020) propose a dispersion reduction attack to make the low-level features featureless. Naseer et al. (2020) design a self-supervised perturbation mechanism for enabling a transferable defense approach. Wu et al. (2020b) compute model attention over extracted features to regularize the search of adversarial examples.

**Generator-oriented Approaches.** Compared with iterative optimization approaches, generator-oriented attacks are more efficient (*i.e.*, only need one inference) to generate adversarial examples. In this branch, Baluja & Fischer (2017) propose an adversarial transformation network to modify the output of the classifier given the original input. Poursaeed et al. (2018) present trainable deep neural networks for producing both image-agnostic and image-dependent perturbations. Naseer et al. (2019) leverage datasets from other domain instead of ImageNet to train generator networks against pre-trained ImageNet models, and inference is performed on ImageNet domain with the aim of fooling black-box ImageNet model. They also attempt a practical black-box threat model (from ChestX-ray to ImageNet), and the attack success rate can outperform the result of Gaussian noise.

## 3 TRANSFERABLE ADVERSARIAL EXAMPLES BEYOND IMAGENET

### 3.1 PROBLEM FORMULATION

Given a target deep learning classifier $f_t(\cdot)$ trained in a specific data distribution $\chi_t$, we aim to craft a human-imperceptible perturbation for the benign image $x_t \sim \chi_t$ from the target domain with the only available knowledge of source ImageNet domain (including pre-trained model $f_s(\cdot)$ and data distribution $\chi_s$). Formally, suppose we have a threat model $\mathcal{M}_{\theta^*}$ whose parameter $\theta^*$ is solely derived from the source domain, our goal is to craft adversarial examples for $x_t$ from target domain so that they can fool the $f_t(\cdot)$ successfully:

$$f_t(\mathcal{M}_{\theta^*}(x_t)) \neq f_t(x_t) \quad s.t. \ ||\mathcal{M}_{\theta^*}(x_t) - x_t||_\infty \leq \epsilon, \tag{1}$$

where $\epsilon$ is the maximum perturbation to ensure $x_t$ is minimally changed. Intuitively, crafting adversarial examples for the black-box domain is very challenging. As shown in Table 1 and Figure 6 of Appendix, images from different domain vary greatly.

### 3.2 PRELIMINARY

Iterative/Single-step optimization methods (Goodfellow et al., 2015; Madry et al., 2018; Zhao et al., 2020; Gao et al., 2021; Mao et al., 2021; Li et al., 2021) and generator-oriented methods (Baluja & Fischer, 2017; Poursaeed et al., 2018; Naseer et al., 2019) are two popular branches for building the threat model. Since the attacker has the large-scale ImageNet training set at hand, there is no reason not to take full advantage of them. Therefore, in this paper, we adopt the generator-oriented framework which learns a transferable adversarial function via a generative model $\mathcal{G}_\theta$. Given that the threat model aims at crafting transferable adversarial examples for black-box domains, relying

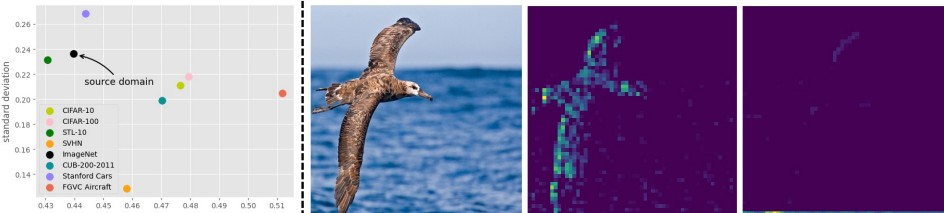

Figure 2: **Left:** The data distribution (*i.e.*, mean and standard deviation) for datasets from different domains. The result is the average over the three channels. **Right:** Two intermediate feature maps (*Maxpool.3*) of VGG-16 (Simonyan & Zisserman, 2015) (trained in ImageNet domain) for the input image from CUB-200-2011 (Wah et al., 2011).

on the last layer with domain-specific loss functions (*e.g.*, relativistic cross-entropy loss adopted by Naseer et al. (2019)) is less effective since this might lead to overfitting to source domain. In contrast, the intermediate layers of the DNN presumably extract general features (Yosinski et al., 2014) which may share across different models. Hence, as a baseline for the new black-domain attack problem, our Beyond ImageNet Attack (BIA) turns to destroy the low-level features of the substitute model at a specific layer $L$ to generate transferable adversarial examples according to existing literature (Yosinski et al., 2014; Zhou et al., 2018; Inkawhich et al., 2019).

As illustrated in Figure 1, $\mathcal{G}_\theta$ is learned to decrease the cosine similarity between adversarial example $\boldsymbol{x'_s}$ and benign image $\boldsymbol{x_s} \in \mathbb{R}^{N \times H_s \times W_s}$ (sampled from $\chi_s$) to make the feature featureless:

$$\theta^* = \arg\min_\theta \mathcal{L}_{cos}(f_s^L(\boldsymbol{x'_s}), f_s^L(\boldsymbol{x_s})). \qquad (2)$$

In the inference phase, our generator $\mathcal{G}_{\theta^*}$ can directly craft adversarial examples for input images $\boldsymbol{x_t} \in \mathbb{R}^{N \times H_t \times W_t}$ from the target domain:

$$\boldsymbol{x'_t} = \min(\boldsymbol{x_t} + \boldsymbol{\epsilon}, \max(\mathcal{G}_{\theta^*}(\boldsymbol{x_t}), \boldsymbol{x_t} - \boldsymbol{\epsilon})). \qquad (3)$$

The resulting adversarial examples $\boldsymbol{x'_t}$ are depicted in Figure 8 of Appendix. Compared with CDA, our BIA is more effective in both source (white-box) and target (black-box) domains. Yet, as shown in Figure 2, crafting more transferable adversarial examples still has some challenges:

- **Data Perspective:** The distribution (*i.e.*, mean and standard deviation) of source domain is largely different from the target domain. For example, the standard deviation of ImageNet is about twice that of SVHN.

- **Model Perspective:** Although some feature map of $f_s^L(\cdot)$ can capture the object ($\in \chi_t$) for feature representation (*e.g.*, the first feature map in Figure 2), there are also some feature maps that are significantly biased (*e.g.*, the second feature map in Figure 2).

To alleviate the concern of generating poor transferable adversarial examples that may arise from the above limitations, we propose two variants, equipped with random normalization ($\mathcal{RN}$) module or domain-agnostic attention ($\mathcal{DA}$) module, respectively.

## 3.3 RANDOM NORMALIZATION MODULE

Generally, DNNs (Simonyan & Zisserman, 2015; He et al., 2016; Huang et al., 2017) are usually equipped with normalization for input images[1] so that they can be modeled as samples from the standard normal distribution. However, as illustrated in Figure 2 (left), the distribution of dataset from the different domain can vary dramatically. Thus, training against a specific domain may limit the generalization of the resulting $\mathcal{G}_{\theta^*}$. Besides, the commonly used strategy of label-preserving data augmentation (Krizhevsky et al., 2012; Simonyan & Zisserman, 2015) is less effective because it has little effect on changing the distribution of the inputs (more details are shown in Appendix A.3).

To overcome this limitation, fusing knowledge from different domains might be helpful. As shown in Table 3 of Naseer et al. (2019), using training data from other domains against the pre-trained

---

[1]For Pytorch pre-trained ImageNet model, input images should be normalized using mean = [0.485, 0.456, 0.406] and standard deviation = [0.229, 0.224, 0.225]

model in the source domain usually enhances the transferability of adversarial examples towards the target domain. However, this setup is not feasible because the available knowledge for a more practical threat model may be limited, *i.e.*, restricted to one domain. Therefore, we instead propose a random normalization ($\mathcal{RN}$) module to simulate different data distribution in the training phase:

$$\mathcal{RN}(\boldsymbol{x_s}) = \boldsymbol{\sigma} \cdot \frac{\boldsymbol{x_s} - \mu'}{\sigma'} + \boldsymbol{\mu}, \tag{4}$$

where $\boldsymbol{\sigma}$ and $\boldsymbol{\mu}$ are default standard deviation and mean vectors for ImageNet, and $\mu' \sim \mathcal{N}(\mu'_{mean}, \mu'_{std})$ and $\sigma' \sim \mathcal{N}(\sigma'_{mean}, \sigma'_{std})$ are two random scales[2] sampled from Gaussian distribution. Combined with $\mathcal{RN}$, and the object function can be expressed as:

$$\theta^* = \arg\min_{\theta} \mathcal{L}_{cos}(f_s^L(\mathcal{RN}(\boldsymbol{x'_s})), f_s^L(\mathcal{RN}(\boldsymbol{x_s}))). \tag{5}$$

### 3.4 DOMAIN-AGNOSTIC ATTENTION MODULE

Unlike the random normalization module, the domain-agnostic attention module aims to narrow the domain gap from the model perspective. Our inspiration is from prior works (Hansen & Salamon, 1990; Caruana et al., 2004; Dong et al., 2018), which demonstrate that the ensemble strategy can avoid getting trapped in the local optimum and improve performance. Since there are many feature maps at layer $L$ and each of them can model the input, *i.e.,* extracts the features, we can also integrate them to produce a more robust feature representation $\mathcal{A}^L$, thus mitigating the impact of several biased feature maps, *e.g.*, the second feature map of Figure 2. Specifically, we apply cross-channel average pooling to the feature maps at layer $L$:

$$\mathcal{A}^L = \frac{|\sum_{i=0}^{C}[f_s^L(\boldsymbol{x_s})]_i|}{C}, \tag{6}$$

where $C$ is channel number of $f_s^L(\boldsymbol{x_s})$.

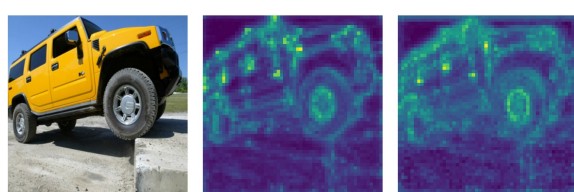

As depicted in Figure 3, even if our source model is not trained in the target domain, the robust feature representation is still able to capture the essential feature of object very well. Surprisingly, it is even similar to the one that derived from a completely different target model. Therefore, this robust feature representation can serve as a domain-agnostic attention ($\mathcal{DA}$) to enhance the cross-domain transferability

Figure 3: **Left:** A benign image from Stanford Cars (Krause et al., 2013). **Middle & Right:** We apply cross-channel average pooling to the intermediate feature maps ($Maxpool.3$) of VGG-16 and ($Conv3\_8$) of DCL (Chen et al., 2019) with backbone Res-50.

of adversarial examples. Specifically, in the training phase, we leverage $\mathcal{A}^L$ to assign weights for each pixel of feature maps at the same layer. The resulting object function can be written as:

$$\theta^* = \arg\min_{\theta} \mathcal{L}_{cos}(\mathcal{A}^L \odot f_s^L(\boldsymbol{x'_s}), \mathcal{A}^L \odot f_s^L(\boldsymbol{x_s})), \tag{7}$$

where $\odot$ is Hadamard product. After the training, even if the inputs are not from the source domain, our generator $\mathcal{G}_{\theta^*}$ is supposed to own an ability to capture the essential feature to disrupt.

## 4 EXPERIMENTS

**Source (white-box) Domain.** Our training data is the large-scale ImageNet (Russakovsky et al., 2015) training set which includes about 1.2 million $224 \times 224 \times 3$ images. Generators are trained against four ImageNet pre-trained models including VGG-16, VGG-19 (Simonyan & Zisserman, 2015), ResNet152 (Res-152) (He et al., 2016) and DenseNet169 (Dense-169) (Huang et al., 2017). In this domain, we also consider three other models, *i.e.*, DenseNet121 (Dense-121) (Huang et al.,

---

[2]$\mu'_{mean}, \mu'_{std}, \sigma'_{mean}$ and $\sigma'_{std}$ are all hyper-parameters.

2017), ResNet50 (Res-50) (He et al., 2016) and Inception-v3 (Inc-v3)[3] (Szegedy et al., 2016), to analyze the cross-model transferability. All models are available in the Torchvision library[4].

**Target (black-box) Domain.** Generally, the image classification tasks can be divided into coarse-grained and fine-grained tasks in terms of label granularity (Touvron et al., 2021). Thus, in addition to ImageNet (white-box domain), we consider seven other black-box domains (shown in Table 1) including four coarse-grained (CIFAR-10, CIFAR-100 (Krizhevsky, 2009), STL-10 (Coates et al., 2011) and SVHN (Netzer et al., 2011)) and three fine-grained (CUB-200-2011 (Wah et al., 2011), Stanford Cars (Krause et al., 2013) and FGVC Aircraft (Maji et al., 2013)) classification tasks. For the fine-grained classification, we use DCL framework (Chen et al., 2019) with three different backbones: ResNet50 (Res-50) (He et al., 2016), SENet154 and SE-ResNet101 (SE-Res101) (Hu et al., 2018). The pre-trained models for the coarse-grained classification are from Github[5].

**Implementation Details.** Our generator $\mathcal{G}_\theta$ adopts the same architecture as (Naseer et al., 2019), which is a composite of downsampling, residual (He et al., 2016) and upsampling blocks (more details can be found in Figure 7 of Appendix). The output size of $\mathcal{G}_\theta$ is equal to the input size. For CDA and our methods, we use Adam optimizer (Kingma & Ba, 2015) with a learning rate of 2e-4 and the exponential decay rate for first and second moments is set to 0.5 and 0.999, respectively. All generators are trained for one epoch with the batch size 16. For the layer $L$, we attack the output of $Maxpool.3$ for VGG-16 and VGG-19, the output of $Conv3\_8$ for Res-152 and the output of $DenseBlock.2$ for Dense-169 (ablation study can be found in Appendix A.1). For brevity, we only refer the output of a specific layer/block using its layer/block name. For our $\mathcal{RN}$ module, we set $\mu' \sim \mathcal{N}(0.50, 0.08)$ and $\sigma' \sim \mathcal{N}(0.75, 0.08)$, and the ablation study is shown in Appendix A.2.

**Competitors.** We compare our proposed methods with projected gradient descent (PGD) (Madry et al., 2018), diverse inputs method (DIM) (Dong et al., 2018; Xie et al., 2019), dispersion reduction (DR) (Lu et al., 2020), self-supervised perturbation (SSP) (Naseer et al., 2020) and cross-domain attack (CDA) (Naseer et al., 2019). The maximum perturbation $\varepsilon$ is set to 10. Follow Lu et al. (2020), we set the step size $\alpha = 4$ and the number of iterations $T = 100$ for all iterative methods. For DIM, we set the default decay factor $\mu = 1.0$ and the transformation probability $p = 0.7$. The Gaussian smoothing (gs) for CDA is applied by $3 \times 3$ Gaussian kernel.

**Evaluation Metrics.** We use the top-1 accuracy on the whole test set (test size is shown in Table 1) after attacking to evaluate the performance of different methods. We also report standard deviation across multiple random runs in Appendix A.7.

## 4.1 TRANSFERABILITY COMPARISONS

In this section, we first conduct experiments for the black-box domain in Section 4.1.1 (coarse-grain) and Section 4.1.2 (fine-grain), then we report the results for the white-box (source) domain in Section 4.1.3. For the discussion of generator mechanism, changing source domain and ensemble-model attacks, we leave them in Appendix A.4, Appendix A.5 and Appendix A.6, respectively.

### 4.1.1 RESULTS ON COARSE-GRAINED DOMAIN

In this section, we craft transferable adversarial examples for coarse-grained classification tasks. The results are shown in Table 2, where we leverage four ImageNet pre-trained models, including VGG-16, VGG-19, Res-152 and Dense-169, to train generators, respectively.

From Table 2, a first glance shows that our proposed methods consistently surpass state-of-the-art approaches. For example, if the substitute model is VGG-16, the most effective CDA remains a top-1 accuracy of 66.41% for CIFAR-10 after attacking, while our vanilla BIA can effectively bring down it to **57.38%** and $\mathcal{DA}$ variant can further drop the top-1 accuracy to **55.16%**. Among all tasks, the STL-10 and SVHN domains are the most difficult to attack, and the performance gap among existing attacks is moderate. Nonetheless, we can significantly enhance the transferability with the help of our $\mathcal{RN}$ variant in these domains. Notably, if the substitute model is Res-152, $\mathcal{RN}$

---

[3]Inc-v3 expects inputs with a size of $299 \times 299 \times 3$. Therefore, the adversarial examples are also craft on the same size inputs. Note the size of training data for our generator is still $224 \times 224 \times 3$.

[4]https://pytorch.org/vision/stable/models.html

[5]https://github.com/aaron-xichen/pytorch-playground

Table 2: Transferability comparisons on four coarse-grained classification tasks. Here we report the top-1 accuracy after attacking (the lower, the better). The generator $\mathcal{G}_\theta$ is trained in the ImageNet domain, and adversarial examples are within the perturbation budget of $\ell_\infty \leq 10$.

| Model | Attacks | CIFAR-10 | CIFAR-100 | STL-10 | SVHN | AVG. | Model | Attacks | CIFAR-10 | CIFAR-100 | STL-10 | SVHN | AVG. |
|---|---|---|---|---|---|---|---|---|---|---|---|---|---|
| | Clean | 93.78 | 74.27 | 77.59 | 96.03 | 85.42 | | Clean | 93.78 | 74.27 | 77.59 | 96.03 | 85.42 |
| VGG-16 | PGD | 79.63 | 48.02 | 74.32 | 94.66 | 74.16 | Res-152 | PGD | 86.17 | 56.38 | 74.51 | 93.94 | 77.75 |
| | DIM | 77.16 | 44.75 | 72.74 | 91.53 | 71.55 | | DIM | 80.50 | 48.03 | 71.2 | 90.87 | 72.65 |
| | DR | 72.49 | 39.04 | 72.56 | 93.27 | 69.34 | | DR | 78.86 | 48.62 | 71.66 | 93.26 | 73.10 |
| | SSP | 68.54 | 33.63 | 72.77 | 93.98 | 67.23 | | SSP | 75.54 | 42.38 | 72.66 | 92.63 | 70.80 |
| | CDA | 66.41 | 32.37 | 72.91 | 92.17 | 65.97 | | CDA | 66.47 | 39.30 | 69.81 | 88.09 | 65.92 |
| | CDA+gs | 86.70 | 59.43 | 73.38 | 91.61 | 77.78 | | CDA+gs | 85.61 | 57.27 | 73.06 | 90.34 | 76.57 |
| | BIA (Ours) | 57.38 | 22.47 | 69.45 | 90.44 | 59.94 | | BIA (Ours) | 65.49 | 33.48 | 69.91 | 89.46 | 64.59 |
| | BIA+$\mathcal{DA}$ (Ours) | 55.16 | 21.71 | 70.00 | 91.76 | 59.66 | | BIA+$\mathcal{DA}$ (Ours) | 65.34 | 32.68 | 69.65 | 91.38 | 64.76 |
| | BIA+$\mathcal{RN}$ (Ours) | **52.81** | **20.82** | **67.55** | **88.03** | **57.30** | | BIA+$\mathcal{RN}$ (Ours) | **61.23** | 32.84 | **68.04** | **85.79** | **61.98** |
| VGG-19 | PGD | 79.15 | 47.73 | 74.71 | 94.86 | 74.11 | Dense-169 | PGD | 84.55 | 54.29 | 74.55 | 93.83 | 76.81 |
| | DIM | 77.54 | 43.81 | 72.96 | 91.68 | 71.50 | | DIM | 80.89 | 49.06 | 72.64 | 89.47 | 73.02 |
| | DR | 70.72 | 37.59 | 71.98 | 93.73 | 68.51 | | DR | 78.24 | 48.67 | 70.75 | 93.2 | 72.72 |
| | SSP | 70.46 | 35.28 | 73.21 | 93.67 | 68.16 | | SSP | 77.13 | 42.18 | 72.53 | 91.64 | 70.87 |
| | CDA | 81.60 | 51.53 | 71.43 | 92.64 | 74.30 | | CDA | 67.75 | 35.03 | 69.00 | 88.76 | 65.14 |
| | CDA+gs | 88.55 | 61.90 | 73.64 | 92.18 | 79.07 | | CDA+gs | 85.01 | 54.71 | 72.61 | 88.69 | 75.26 |
| | BIA (Ours) | 57.88 | 23.12 | 69.84 | 88.89 | 59.93 | | BIA (Ours) | 72.02 | 38.99 | 69.80 | 86.12 | 66.73 |
| | BIA+$\mathcal{DA}$ (Ours) | 57.26 | 23.04 | 70.16 | 90.08 | 60.14 | | BIA+$\mathcal{DA}$ (Ours) | 71.69 | 38.95 | 70.60 | 88.02 | 67.32 |
| | BIA+$\mathcal{RN}$ (Ours) | **54.47** | **22.61** | **68.23** | **88.08** | **58.35** | | BIA+$\mathcal{RN}$ (Ours) | **66.67** | **34.41** | **68.79** | **81.54** | **62.85** |

Table 3: Transferability comparisons on three fine-grained classification tasks. Here we report the top-1 accuracy after attacking (the lower, the better). The generator $\mathcal{G}_\theta$ is trained in ImageNet domain and adversarial examples are within the perturbation budget of $\ell_\infty \leq 10$.

| Model | Attacks | CUB-200-2011 | | | Stanford Cars | | | FGVC Aircraft | | | AVG. |
|---|---|---|---|---|---|---|---|---|---|---|---|
| | | Res-50 | SENet154 | SE-Res101 | Res-50 | SENet154 | SE-Res101 | Res-50 | SENet154 | SE-Res101 | |
| | Clean | 87.35 | 86.81 | 86.56 | 94.35 | 93.36 | 92.97 | 92.23 | 92.08 | 91.90 | 90.85 |
| VGG-16 | PGD | 80.65 | 79.58 | 80.69 | 87.45 | 89.04 | 90.30 | 84.88 | 83.92 | 82.15 | 84.30 |
| | DIM | 70.02 | 62.86 | 70.57 | 74.72 | 78.10 | 84.33 | 73.54 | 66.88 | 62.38 | 71.49 |
| | DR | 81.08 | 82.05 | 82.52 | 90.82 | 90.59 | 91.12 | 84.97 | 87.55 | 85.54 | 86.25 |
| | SSP | 62.27 | 60.44 | 71.52 | 58.02 | 75.71 | 83.02 | 54.91 | 68.74 | 63.79 | 66.49 |
| | CDA | 69.69 | 62.51 | 71.00 | 75.94 | 72.45 | 84.64 | 71.53 | 58.33 | 63.39 | 69.94 |
| | CDA+gs | 70.19 | 63.19 | 68.92 | 85.03 | 79.52 | 83.52 | 78.55 | 65.62 | 68.38 | 73.66 |
| | BIA (Ours) | 32.74 | 52.99 | 58.04 | 39.61 | 69.90 | 70.17 | 28.92 | 60.31 | 46.92 | 51.07 |
| | BIA+$\mathcal{DA}$ (Ours) | **25.00** | **40.27** | **53.24** | 22.24 | **59.48** | **61.52** | **15.36** | **47.91** | **40.83** | **40.65** |
| | BIA+$\mathcal{RN}$ (Ours) | 26.13 | 46.15 | 55.07 | **20.61** | 62.64 | 63.38 | 16.50 | 52.54 | 45.48 | 43.17 |
| VGG-19 | PGD | 80.98 | 79.00 | 80.60 | 87.54 | 88.87 | 90.56 | 84.70 | 84.01 | 83.35 | 84.40 |
| | DIM | 69.93 | 61.60 | 70.90 | 75.02 | 78.55 | 84.63 | 74.59 | 67.69 | 65.26 | 72.02 |
| | DR | 80.83 | 81.57 | 81.95 | 91.00 | 90.23 | 91.15 | 84.43 | 85.96 | 84.57 | 85.74 |
| | SSP | 62.94 | 58.34 | 70.45 | 61.90 | 76.27 | 83.77 | 58.78 | 69.52 | 66.88 | 67.65 |
| | CDA | 59.48 | 61.08 | 68.50 | 58.53 | 70.70 | 80.70 | 59.26 | 52.24 | 62.26 | 63.64 |
| | CDA+gs | 67.88 | 59.42 | 67.57 | 82.83 | 78.61 | 82.64 | 79.36 | 65.89 | 68.35 | 72.51 |
| | BIA (Ours) | 48.90 | 52.33 | 56.47 | 66.34 | 72.45 | 75.08 | 50.95 | 54.04 | 51.79 | 58.71 |
| | BIA+$\mathcal{DA}$ (Ours) | **27.46** | **37.61** | **50.14** | **35.27** | **61.40** | **64.41** | **17.97** | **45.81** | **44.01** | **42.68** |
| | BIA+$\mathcal{RN}$ (Ours) | 31.77 | 43.41 | 51.09 | 42.81 | 68.90 | 66.27 | 27.75 | 52.48 | 45.57 | 47.78 |
| Res-152 | PGD | 73.21 | 75.89 | 76.11 | 83.99 | 86.89 | 88.24 | 79.00 | 79.30 | 75.64 | 79.81 |
| | DIM | 56.30 | 59.35 | 63.36 | 67.88 | 76.37 | 79.77 | 64.21 | 62.95 | 54.49 | 64.96 |
| | DR | 77.58 | 82.07 | 80.60 | 87.96 | 90.11 | 90.67 | 77.89 | 82.27 | 80.08 | 83.25 |
| | SSP | 47.64 | 66.17 | 67.90 | 53.29 | 79.09 | 83.45 | 58.35 | 73.36 | 69.34 | 66.51 |
| | CDA | 45.15 | 53.69 | 52.86 | 57.72 | 63.09 | 73.05 | 64.87 | 46.74 | 59.11 | 57.36 |
| | CDA+gs | 59.32 | 64.24 | 60.08 | 81.15 | 82.91 | 82.56 | 75.52 | 73.72 | 66.58 | 71.79 |
| | BIA (Ours) | 43.55 | 49.50 | 55.54 | 31.65 | 54.12 | 67.21 | 38.49 | 32.46 | 51.19 | 47.08 |
| | BIA+$\mathcal{DA}$ (Ours) | 25.80 | **39.71** | 52.83 | 33.43 | 52.41 | 68.03 | **27.51** | **27.42** | 47.28 | 41.60 |
| | BIA+$\mathcal{RN}$ (Ours) | **23.54** | 40.13 | **51.36** | **12.39** | **47.92** | **60.59** | 32.34 | 32.85 | **46.89** | **38.67** |
| Dense-169 | PGD | 79.29 | 81.14 | 79.74 | 87.66 | 90.13 | 89.80 | 83.23 | 84.31 | 81.24 | 84.06 |
| | DIM | 63.17 | 62.01 | 65.96 | 72.88 | 78.29 | 81.25 | 68.89 | 65.26 | 54.79 | 68.06 |
| | DR | 74.84 | 78.89 | 77.93 | 86.54 | 88.82 | 89.52 | 77.80 | 78.73 | 74.47 | 80.84 |
| | SSP | 41.80 | 49.95 | 59.72 | 26.65 | 68.71 | 74.36 | 16.80 | 55.78 | 44.07 | 48.65 |
| | CDA | 52.92 | 60.96 | 57.04 | 53.64 | 73.66 | 75.51 | 62.23 | 61.42 | 59.83 | 61.91 |
| | CDA+gs | 60.86 | 61.34 | 60.10 | 74.95 | 76.35 | 78.86 | 72.94 | 68.68 | 64.66 | 68.75 |
| | BIA (Ours) | 21.79 | **29.29** | 39.13 | 9.58 | 44.46 | 49.06 | 8.04 | 27.84 | 33.87 | 29.23 |
| | BIA+$\mathcal{DA}$ (Ours) | 12.36 | 29.31 | **35.78** | **4.56** | **40.85** | **31.82** | 3.90 | **13.59** | **14.55** | **20.75** |
| | BIA+$\mathcal{RN}$ (Ours) | **10.67** | 32.50 | 46.62 | 7.76 | 44.14 | 42.76 | **3.81** | 34.20 | 27.00 | 27.72 |

variant can further decrease the top-1 accuracy from 89.46% (vanilla BIA) to **85.79%** on SVHN. Compared with state-of-the-art CDA on all domains, $\mathcal{RN}$ variant significantly outperforms it by **7.71%** on average. This demonstrates that our proposed $\mathcal{RN}$ module is effective in coping with different distributions of inputs, thus improving the generalization of the resulting generator.

### 4.1.2 RESULTS ON FINE-GRAINED DOMAIN

We also analyze the transferability of adversarial examples towards fine-grained classification tasks. For each domain, three black-box models with different backbones trained via the DCL framework are the target. The results are summarized in Table 3, where the leftmost column is the substitute model and the top row shows the target model.

In this scenario, the performance gap between the existing state-of-the-art algorithms and our proposed methods is further enlarged. Remarkably, when attacking against Dense-169, even the vanilla BIA can drop the average top-1 accuracy to **29.23%**, while CDA+gs, CDA, SSP, DR, DIM and PGD are still with the high average top-1 accuracy of 68.75%, 61.91%, 48.65%, 80.84%, 68.06% and 84.06% after attacking, respectively. Furthermore, by adding $\mathcal{DA}$ or $\mathcal{RN}$ modules in the training phase, the generator $\mathcal{G}_{\theta*}$ is capable of crafting more transferable adversarial examples. On average, our $\mathcal{RN}$ variant can reduce the top-1 accuracy from 46.52% (vanilla BIA) to **39.33%**, and $\mathcal{DA}$ variant can further drop it to **36.42%**, which remarkably outperforms SSP by **25.91%**. This demonstrates our proposed $\mathcal{DA}$ module can effectively alleviate the bias caused by several feature maps, thus focusing on disrupting essential features.

Table 4: Transferability comparisons on ImageNet (source domain). Here we report the top-1 accuracy after attacking (the lower, the better). The generator $\mathcal{G}_{\theta}$ is trained in ImageNet domain ("*" denotes white-box model) and adversarial examples are within the perturbation budget of $\ell_{\infty} \leq 10$.

| Model | Attack | VGG-16 | Dense-169 | VGG-19 | Res-50 | Res-152 | Dense-121 | Inc-v3 | AVG. |
|---|---|---|---|---|---|---|---|---|---|
| Model | Clean | 70.14 | 75.75 | 70.95 | 74.61 | 77.34 | 74.22 | 76.19 | 74.17 |
| | PGD | 2.49* | 53.22 | 4.27 | 45.52 | 58.69 | 48.28 | 61.08 | 39.08 |
| | DIM | 3.32* | **22.72** | 3.39 | 19.13 | 33.03 | 18.50 | 30.05 | 18.59 |
| | DR | 20.95* | 67.91 | 43.59 | 64.90 | 70.00 | 64.73 | 69.22 | 57.33 |
| | SSP | 0.95* | 39.56 | 3.42 | 26.22 | 41.68 | 34.45 | 47.46 | 27.68 |
| VGG-16 | CDA | **0.40*** | 42.67 | **0.77** | 36.27 | 51.05 | 38.89 | 54.02 | 32.01 |
| | CDA+gs | 12.10* | 57.09 | 20.48 | 51.87 | 60.83 | 52.21 | 55.12 | 44.24 |
| | BIA (Ours) | 1.55* | 32.35 | 3.61 | 25.36 | 42.98 | 26.97 | 41.20 | 24.86 |
| | BIA+$\mathcal{DA}$ (Ours) | 1.04* | 24.52 | 2.07 | 18.63 | 36.43 | 19.97 | 34.54 | 19.60 |
| | BIA+$\mathcal{RN}$ (Ours) | 1.44* | 25.96 | 2.58 | **16.52** | **31.80** | 18.25 | **28.54** | **17.87** |
| | PGD | 38.51 | 5.03* | 40.53 | 33.91 | 44.97 | 21.18 | 58.30 | 34.63 |
| | DIM | 12.31 | 5.25* | 13.20 | 8.98 | 12.93 | 5.91 | 21.44 | 11.43 |
| | DR | 38.45 | 23.99* | 41.59 | 50.19 | 58.70 | 49.95 | 63.70 | 46.65 |
| | SSP | 11.53 | 1.32* | 12.54 | 12.97 | 25.66 | 9.74 | 25.58 | 14.19 |
| Dense-169 | CDA | 7.26 | **0.63*** | 7.91 | 6.46 | 15.56 | 5.13 | 43.78 | 12.39 |
| | CDA+gs | 27.98 | 22.95* | 28.56 | 31.52 | 43.67 | 31.66 | 49.18 | 33.65 |
| | BIA (Ours) | 4.76 | 6.45* | 7.15 | 6.97 | 13.83 | 6.60 | 38.58 | 12.05 |
| | BIA+$\mathcal{DA}$ (Ours) | **3.17** | 3.32* | **4.09** | **4.44** | **5.85** | **3.98** | 26.51 | **7.34** |
| | BIA+$\mathcal{RN}$ (Ours) | 3.66 | 4.05* | 5.23 | 6.91 | 13.25 | 4.21 | **14.24** | 7.36 |

### 4.1.3 RESULTS ON SOURCE DOMAIN

Although our proposed methods are mainly designed to improve the threat of adversarial examples towards black-box domains, they are also effective for enhancing the cross-model black-box transferability in the white-box domain. For example, by training against Dense-169, CDA still remains a top-1 accuracy of 43.78% on Inc-v3, while our BIA can achieve a relatively low top-1 accuracy of 38.58% on it. Besides, $\mathcal{RN}$ variant can further decrease the top-1 accuracy on Dense-169 (white-box model) and Inc-v3 (black-box model) by **2.4%** and **24.34%**, respectively. This demonstrates that our proposed $\mathcal{RN}$ module is also able to avoid getting stuck in the local optimum of a specific model when training on a large-scale dataset. For $\mathcal{DA}$ variant, since the target domain and source domain are identical, it is naturally able to improve the transferability as well. As shown in Table 4, compared with vanilla BIA, $\mathcal{DA}$ variant can further degrade the top-1 accuracy from 18.45% (vanilla BIA) to **13.47%** on average.

### 4.2 COMBINATION OF DOMAIN-AGNOSTIC ATTENTION AND RANDOM NORMALIZATION

In this section, we report the results for BIA equipped with both $\mathcal{RN}$ and $\mathcal{DA}$. The following is the update rule:

$$\theta^* = \arg\min_{\theta} \mathcal{L}_{cos}(\mathcal{A}^L \odot f_s^L(\mathcal{RN}(\boldsymbol{x_s'})), \mathcal{A}^L \odot f_s^L(\mathcal{RN}(\boldsymbol{x_s}))). \tag{8}$$

As illustrated in Figure 4, $\mathcal{RN}$ module and $\mathcal{DA}$ module are not always mutually reinforcing. For example, when training against Res-152 and transferring adversarial examples to fine-grained or source domains, combining $\mathcal{RN}$ module and $\mathcal{DA}$ module can further decrease the average top-1 accuracy to 35.75% and 13.11%, respectively. However, if the black-box domain is coarse-grain,

only applying $\mathcal{RN}$ module is better than applying both $\mathcal{RN}$ and $\mathcal{DA}$ modules. We speculate that it may be because the $\mathcal{RN}$ module affects the low-level features extract by the substitute model, and thus be incompatible with the $\mathcal{DA}$ module sometimes.

Since Figure 4 shows that using $\mathcal{DA}$ and $\mathcal{RN}$ in tandem is less effective when training against Dense-169 while they reinforce each other in VGG-16 in most cases, we visualize the cross-channel average pooling of intermediate features for VGG-16 and Dense-169 to better explain this phenomenon. As illustrated in Figure 5, it can be observed that the $\mathcal{RN}$ module reinforces the discriminative features in VGG-16. However, it in-

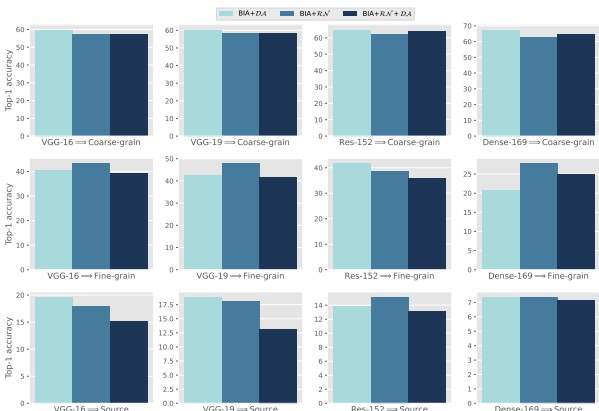

Figure 4: The average top-1 accuracy of $\mathcal{DA}$, $\mathcal{RN}$ and $\mathcal{DA} + \mathcal{RN}$ variants after attacking on coarse-grained, fine-grained and source domains.

hibits the response of objects' essential features extracted by Dense-169. Consequently, $\mathcal{DA}$ module may cause the resulting generator to reduce the ability to attack essential features, thereby making it challenging to use these two techniques in tandem.

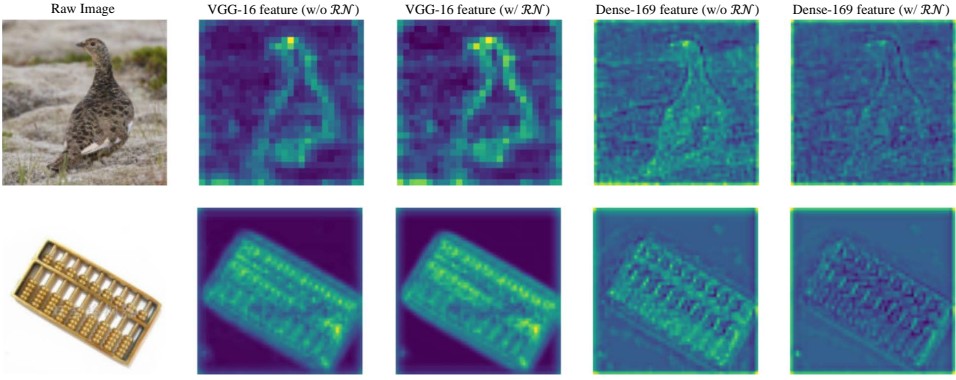

Figure 5: Visualization (cross-channel average pooling) of intermediate features for VGG-16 and Dense-169. It can be observed that the $\mathcal{RN}$ module inhibits the response of objects' essential feature extracted by Dense-169, while VGG-16 enhances the response.

## 5 CONCLUSION

In this paper, we present a practical black-box threat model for the cross-domain attack. Specifically, we train a generator network in the large-scale ImageNet domain to disrupt low-level features better, thus generating transferable adversarial examples for the black-box domain. Based on this framework, we further propose two variants to narrow the gap between the source and target domains from the data and model perspectives, respectively. Extensive experiments demonstrate the effectiveness of our proposed methods. This also reminds the model owner that "Your deployed model is not safe even you do not leak any information to the public". We hope our proposed approaches can serve as a benchmark for evaluating the stability of various deployed models.

## 6 ACKNOWLEDGE

This work was supported by the National Natural Science Foundation of China (Grant No. 62020106008, No. 61772116 and No. 61872064) and Alibaba Group.

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

# A   APPENDIX

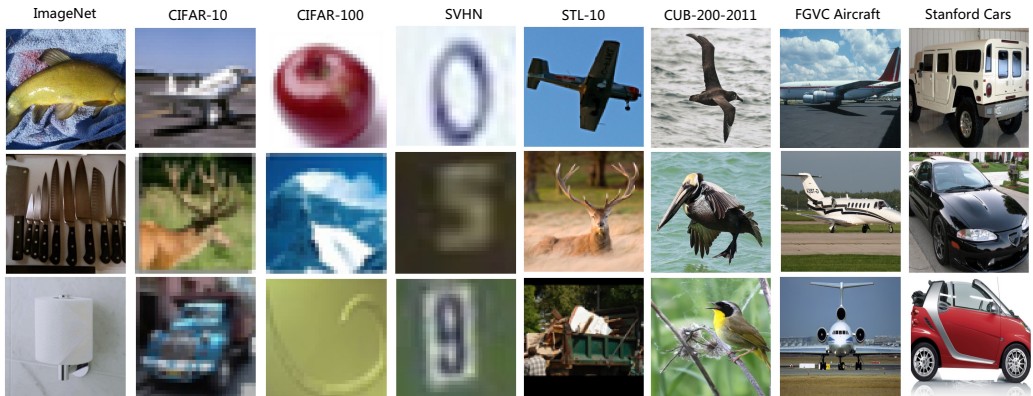

Figure 6: Benign images sampled from each domain. From the top to the bottom rows are the first category, the middle category and the last category of their label space, respectively.

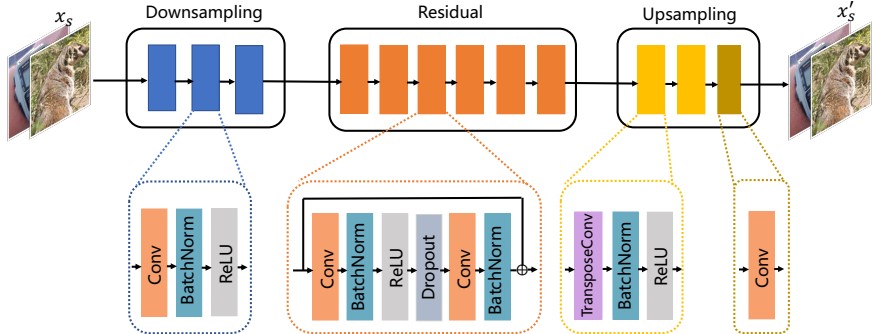

Figure 7: The structure of the generator.

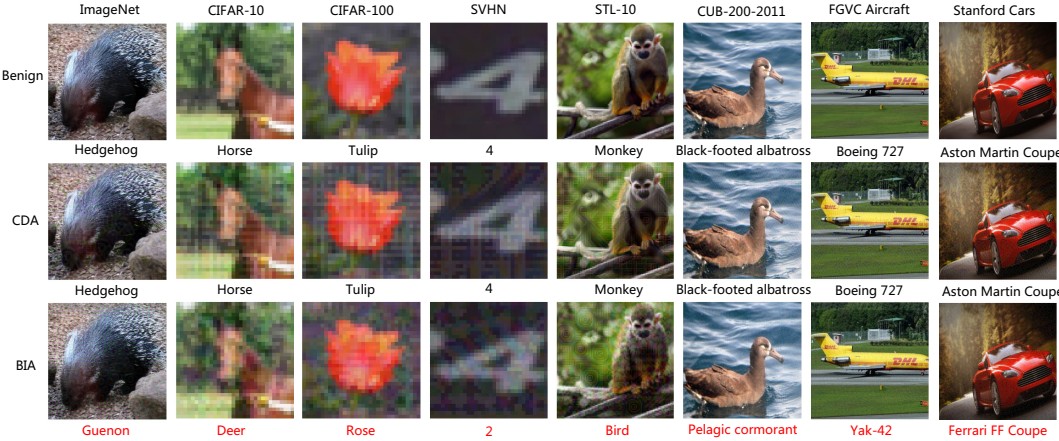

Figure 8: Adversarial examples crafted by CDA and our vanilla BIA ($\epsilon = 10$). Both generator networks are trained against ImageNet pre-trained VGG-16 (Simonyan & Zisserman, 2015). Red highlighted labels represent misclassification.

## A.1 SELECT LAYER FOR ATTACKING

In this section, we analyze the impact of different intermediate layers of the substitute model on the transferability of resulting adversarial examples. The results are illustrated in Figure 10.

In general, training against the shallow and middle layers yields more cross-domain transferable but less cross-model transferable adversarial examples than against the deep layer. For example, if the substitute model is Res-152, disrupting shallow layer $Conv2\_3$ is more effective for transferring towards coarse-grained domain, and perturbing middle layer $Conv3\_8$ is more effective in reducing the average top-1 accuracy of fine-grained models. In contrast, attacking deep layers like $Conv5\_3$ can yield more transferable adversarial examples in the source domain. This demonstrates that low-level features are more similar across domains and high-level features are more domain-specific.

## A.2 SELECT GAUSSIAN DISTRIBUTION FOR RANDOM NORMALIZATION

For $\mathcal{DA}$ module, it is parameter-free. Therefore, we only conduct the experiment to select an optimal distribution for $\mathcal{RN}$ module, *i.e.*, the $\mu'$ and $\sigma'$ in Equation 4. Here we tune the mean of $\mu'$ and $\sigma'$ from 0.25 to 0.75 with a granularity of 0.25. For the standard deviation of them, we fix it to 0.08 so that the sampled $\mu'$ and $\sigma'$ can basically take values ranging from 0.0 to 1.0 (according to the three-sigma rule).

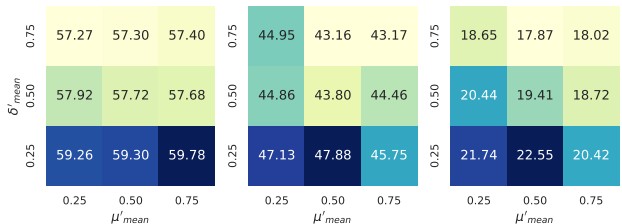

Figure 9: The average top-1 accuracy after attacking on coarse-grained (left), fine-grained (middle) and source (right) domains with different $\mu'_{mean}$ and $\sigma'_{mean}$ for $\mathcal{RN}$.

The results are shown in Figure 9, where we craft adversarial examples via VGG-16 and report the average top-1 accuracy in both source and black-box domains. From the results, we observe that a bigger $\sigma'_{mean}$ can effectively improve the transferability. For example, if we increase $\sigma'_{mean}$ from 0.25 to 0.75, the top-1 accuracy on fine-grained models can be further decreased by 3.16% on average. Although the influence of $\mu'_{mean}$ is relatively moderate when $\sigma'_{mean} = 0.75$, setting $\mu'_{mean}$ to 0.5 is usually better. Therefore, we set $\mu' \sim \mathcal{N}(0.50, 0.08)$ and $\sigma' \sim \mathcal{N}(0.75, 0.08)$ in our paper.

## A.3 DATA AUGMENTATION VS. RANDOM NORMALIZATION

Data augmentation (Krizhevsky et al., 2012; Simonyan & Zisserman, 2015) is a widely used strategy for improving the generalization of the model. Nonetheless, these label-preserving transformations are less effective for training a generator to craft more transferable adversarial examples.

In Figure 11, we report the results for our vanilla BIA, BIA with data augmentation (BIA+AUG)[6] and BIA with $\mathcal{RN}$ (BIA+$\mathcal{RN}$). As shown in Figure 11, BIA+AUG is less effective than our proposed BIA+$\mathcal{RN}$ and might even degrade the performance of our vanilla BIA. For example, when training against VGG-19, BIA gets an average top-1 accuracy of 58.71% on the fine-grained domain, yet BIA+AUG degrades it to 62.29%. In contrast, our proposed BIA+$\mathcal{RN}$ can significantly decrease the result to **47.78%**. This is mainly because that the common data augmentation cannot effectively change the distribution of the training dataset, thus decreasing the generalization towards the black-box domain.

## A.4 INSIGHT INTO THE GENERATOR

Although many prior works (Poursaeed et al., 2018; Naseer et al., 2019) have leveraged the generator to craft adversarial examples, they hardly analyze the role of each block of the generator. This section will give an insight into the generator and understand how it processes an input. In Figure 12, we feed an image into the generator trained with the vanilla BIA (against Res-152) and visualize the

---

[6]Specifically, we use three common data augmentation techniques including "RandomResizedCrop", "RandomHorizontalFlip" and "ColorJitter".

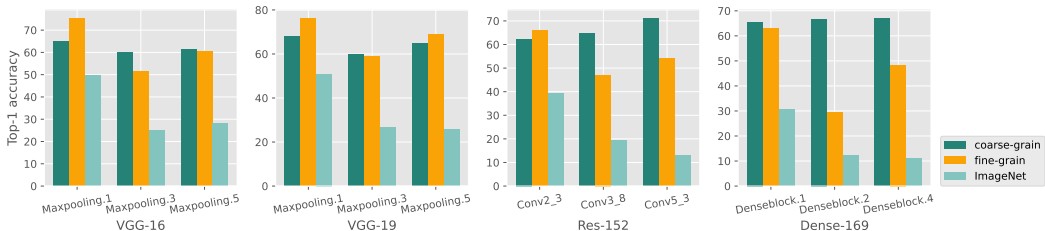

Figure 10: The average top-1 accuracy after attacking (the lower, the better) on coarse-grained, fine-grained and source domains. Our generator $\mathcal{G}_\theta$ is trained against different layers (from shallow to deep) of ImageNet pre-trained VGG-16, VGG-19, Res-152 and Dense-169, respectively.

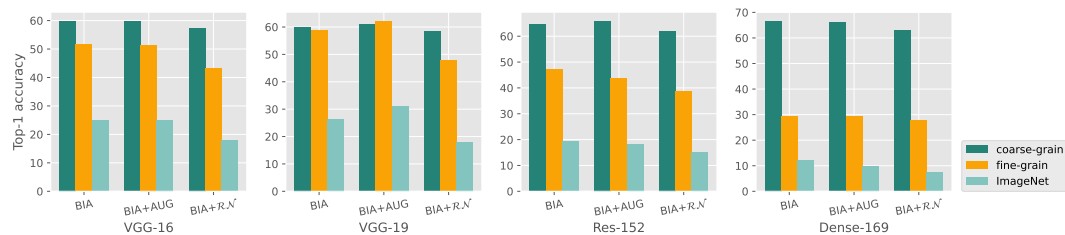

Figure 11: The average top-1 accuracy after attacking on coarse-grained, fine-grained and source domains. Here we compare the results of vanilla BIA, BIA with data augmentation (AUG) and BIA with $\mathcal{RN}$. Our generator $\mathcal{G}_\theta$ is trained against ImageNet pre-trained VGG-16, VGG-19, Res-152 and Dense-169, respectively.

output of each block. As we can observe, these blocks play different roles in crafting adversarial examples:

- Downsampling block: it mainly extracts discriminative features of the input image. As the size of the down-sampled images gets smaller, there is no significant noise overall.

- Residual block: Unlike the downsampling block, this block is responsible for adding noise and its behavior looks very similar to the iterative algorithm.

- Upsampling block: This block gradually reconstructs the adversarial example from abstract features.

Since the residual block mainly works as suppressing or reversing the features extracted by the downsampling module, the downsampling module has an essential impact on generating transferable adversarial examples.

To better understand the effectiveness of our proposed modules, here we investigate them from the perspective of the generator mechanism. Specifically, we apply cross-channel average pooling to the output of the downsampling block and calculate the difference map between vanilla BIA and our proposed variants. Without loss of generality, here we only show the definition of $Diff(\mathcal{RN}\ variant$, BIA), and $Diff(\mathcal{DA}\ variant$, BIA) can be easily deduced. Specifically, we first apply cross-channel average pooling to the output of the downsampling

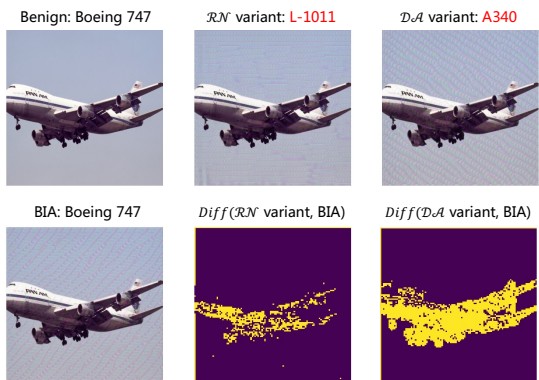

Figure 13: A benign image (label is "Boeing 747") from FGVC Aircraft (Maji et al., 2013) and its corresponding adversarial examples and difference map.

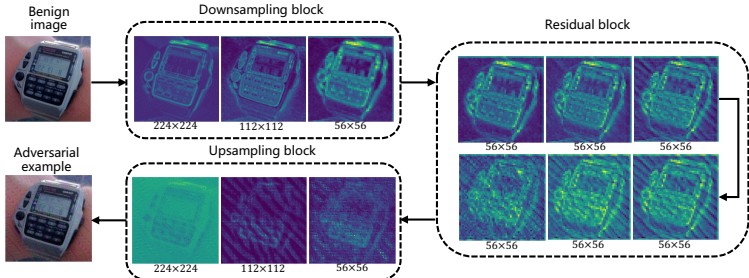

Figure 12: We apply cross-channel average pooling to visualize each block of our generator $\mathcal{G}_{\theta^*}$.

block:

$$\mathcal{A}_{RN}^L = \frac{|\sum_{i=0}^C [\mathcal{G}_{\mathcal{RN}_{\theta^*}}^d(\boldsymbol{x})]_i|}{C}, \tag{9}$$

$$\mathcal{A}_{BIA}^L = \frac{|\sum_{i=0}^C [\mathcal{G}_{BIA_{\theta^*}}^d(\boldsymbol{x})]_i|}{C}, \tag{10}$$

where $\mathcal{G}_{\mathcal{RN}_{\theta^*}}^d$ and $\mathcal{G}_{\mathcal{BIA}_{\theta^*}}^d$ denote the output of the downsampling block for $\mathcal{RN}$ variant and vanilla BIA, respectively. Then our difference map can be expressed by:

$$Diff(\mathcal{RN}\ variant, BIA) = \begin{cases} 1, & \mathcal{A}_{RN}^L - \mathcal{A}_{BIA}^L > 0, \\ 0, & else. \end{cases} \tag{11}$$

From the result of Figure 13, we can observe that the generators derived from our proposed variants concentrate more on the body of the object (especially for $\mathcal{DA}$ variant) than that of vanilla BIA (which pays more attention to the background, *i.e.*, the black region in the difference map). This demonstrates that our proposed $\mathcal{RN}$ and $\mathcal{DA}$ variants do narrow the domain gap, and thus be capable of yielding more transferable adversarial examples.

## A.5   DISCUSSION ON CHANGING SOURCE DOMAIN

Since all the experiments are only regarding from ImageNet domains to target domains, it is unclear how well the method will perform if the source dataset is different (especially when the source dataset is small). Therefore, in the following Table 5, we report the results for transferability from CUB-200-2011 to other domains. Generators are learned against CUB-200-2011 domain (the substitute model is DCL (backbone: Res-50) and training data is CUB-200-2011 testing data (5794 images)). For results of fine-grained domains, we average top-1 accuracy of backbone SENet-154 and SE-Res101. For the result of the ImageNet domain, we average top-1 accuracy of all models introduced in our manuscript. We can observe that our method consistently outperforms our main competitor CDA by a large margin. Besides, we also notice that transferring from CUB-200-2011 to CIFAR is very challenging (compared with our reported results in Table 2). Therefore, we highlight the necessity of using a large-scale dataset such as ImageNet to train the adversarial examples generator.

Table 5: Transferability comparisons of CDA and our methods. Here we report the top-1 accuracy after attacking (the lower, the better). The generator $\mathcal{G}_\theta$ is trained in CUB-200-2011 domain and adversarial examples are within the perturbation budget of $\ell_\infty \leq 10$.

| Attacks | CUB-200-2011 | CIFAR-10 | CIFAR-100 | STL-10 | SVHN | Stanford Cars | FGVC Aircraft | ImageNet |
|---|---|---|---|---|---|---|---|---|
| CDA | 64.34 | 83.61 | 54.83 | 70.49 | 91.81 | 74.37 | 71.17 | 50.99 |
| BIA | **40.40** | **82.62** | 54.57 | 71.43 | 91.67 | 68.25 | 57.90 | 44.15 |
| BIA+$\mathcal{DA}$ | **29.55** | 82.94 | 54.15 | **71.70** | **87.28** | **54.65** | 55.15 | **37.15** |
| BIA+$\mathcal{RN}$ | 42.88 | **83.84** | **53.63** | 69.79 | 87.95 | 57.01 | **50.85** | 39.43 |

A.6 DISCUSSION ON ENSEMBLE-MODEL ATTACKS

As demonstrated in prior works (Liu et al., 2017; Dong et al., 2018; Mopuri et al., 2018), attacking against an ensemble of models can yield more transferable adversarial examples. However, it is not clear whether the ensemble-model attack is also effective in our cross-domain attack scenario. To investigate this, we conduct an experiment in Table 6, which shows the results for training against VGG-16 and an ensemble of VGG-16, Vgg-19, Res-152 and Dense-169, respectively.

From the result, We can observe that ensemble-based training can also improve the transferability of adversarial examples towards black-box domains significantly.

Table 6: Transferability comparisons of singe-model (i.e. VGG-16) attacks and ensemble-model (i.e. an ensemble of VGG-16, VGG-19, Res-152 and Dense-169) attacks. Here we report the top-1 accuracy after attacking (the lower, the better) and adversarial examples are within the perturbation budget of $\ell_\infty \leq 10$.

| | CIFAR-10 | CIFAR-100 | STL-10 | SVHN | CUB-200-2011 (SENet-154) | Stanford Cars (SENet-154) | FGVC Aircraft (SENet-154) |
|---|---|---|---|---|---|---|---|
| BIA | 57.38 | 22.47 | 69.45 | 90.44 | 52.99 | 69.90 | 60.31 |
| BIA (ensemble) | **54.96** | **21.73** | **68.94** | **85.85** | **29.15** | **46.47** | **36.87** |
| BIA+$\mathcal{DA}$ | 55.16 | 21.71 | 70.00 | 91.76 | 40.27 | 59.48 | 47.91 |
| BIA+$\mathcal{DA}$ (ensemble) | **52.97** | **20.51** | **69.51** | **89.15** | **18.47** | **38.99** | **25.41** |
| BIA+$\mathcal{RN}$ | 52.81 | **20.82** | 67.55 | 88.03 | 46.15 | 62.64 | 52.54 |
| BIA+$\mathcal{RN}$ (ensemble) | **50.99** | 22.35 | **66.06** | **81.66** | **35.40** | **41.87** | **27.81** |

A.7 DISCUSSION ON STANDARD DEVIATION ACROSS MULTIPLE RANDOM RUNS

To ensure the stability and credibility of the evaluations, experiments are repeated several times for our methods. In Table 7, we report the results for each random seed and standard deviation across these random runs.

Table 7: We show the results of 5 random seeds for methods. Here we report the top-1 accuracy after attacking (the lower, the better). The generator $\mathcal{G}_\theta$ is trained against VGG-16 and adversarial examples are within the perturbation budget of $\ell_\infty \leq 10$.

| | | CIFAR-10 | CIFAR-100 | STL-10 | SVHN | CUB-200-2011 (SENet-154) | Stanford Cars (SENet-154) | FGVC Aircraft (SENet-154) | ImageNet (Res-152) |
|---|---|---|---|---|---|---|---|---|---|
| BIA | Paper report (Table 2 & 3 & 4) | 57.38 | 22.47 | 69.45 | 90.44 | 52.99 | 69.90 | 60.31 | 42.98 |
| | Random runs | 56.75 | 21.86 | 69.61 | 90.48 | 52.47 | 68.13 | 61.45 | 44.04 |
| | | 57.31 | 22.32 | 69.83 | 90.35 | 52.04 | 69.48 | 57.37 | 41.67 |
| | | 57.13 | 22.82 | 70.06 | 90.01 | 53.27 | 69.74 | 58.54 | 41.02 |
| | | 57.03 | 22.38 | 70.05 | 90.43 | 53.04 | 71.91 | 62.02 | 43.09 |
| | | 57.36 | 22.50 | 69.55 | 89.78 | 51.48 | 68.15 | 58.36 | 42.00 |
| | Result | 57.16 ± 0.24 | 22.39 ± 0.31 | 69.76 ± 0.26 | 90.25 ± 0.29 | 52.55 ± 0.69 | 69.55 ± 1.39 | 59.68 ± 1.86 | 42.47 ± 1.10 |
| BIA+$\mathcal{RN}$ | Paper report (Table 2 & 3 & 4) | 52.81 | 20.82 | 67.55 | 88.03 | 46.15 | 62.64 | 52.54 | 31.80 |
| | Random runs | 52.56 | 21.12 | 66.75 | 88.63 | 44.84 | 63.18 | 51.55 | 30.71 |
| | | 52.63 | 21.43 | 67.24 | 87.93 | 48.38 | 63.4 | 55.09 | 33.76 |
| | | 52.29 | 21.42 | 67.3 | 88.64 | 45.41 | 63.2 | 52.99 | 31.8 |
| | | 52.60 | 21.68 | 67.21 | 87.69 | 48.88 | 63.98 | 52.69 | 29.56 |
| | | 53.41 | 21.66 | 67.00 | 88.40 | 49.01 | 64.16 | 55.96 | 30.96 |
| | Result | 52.70 ± 0.42 | 21.46 ± 0.23 | 67.10 ± 0.23 | 88.26 ± 0.43 | 47.30 ± 2.01 | 63.58 ± 0.46 | 53.66 ± 1.81 | 31.36 ± 1.56 |
| BIA+$\mathcal{DA}$ | Paper report (Table 2 & 3 & 4) | 55.16 | 21.71 | 70.00 | 91.76 | 40.27 | 59.48 | 47.91 | 36.43 |
| | Random runs | 55.05 | 21.8 | 69.93 | 91.43 | 41.01 | 57.70 | 48.15 | 35.81 |
| | | 54.47 | 21.34 | 70.05 | 91.5 | 39.96 | 57.98 | 48.30 | 35.23 |
| | | 55.30 | 21.37 | 69.79 | 92.15 | 41.89 | 58.65 | 48.78 | 36.73 |
| | | 55.15 | 21.31 | 70.05 | 92.18 | 40.73 | 59.15 | 49.08 | 36.09 |
| | | 54.52 | 21.36 | 69.71 | 92.04 | 41.92 | 58.57 | 49.02 | 37.67 |
| | Result | 54.90 ± 0.38 | 21.44 ± 0.20 | 69.91 ± 0.15 | 91.86 ± 0.37 | 41.10 ± 0.83 | 58.41 ± 0.57 | 48.67 ± 0.42 | 36.31 ± 0.93 |
| BIA+$\mathcal{DA}$+$\mathcal{RN}$ | Paper report (Table 2 & 3 & 4) | 50.29 | 20.03 | 67.51 | 90.4 | 40.46 | 57.26 | 37.01 | 24.91 |
| | Random runs | 51.35 | 20.18 | 67.53 | 91.34 | 43.56 | 58.29 | 36.11 | 22.25 |
| | | 50.49 | 20.17 | 67.5 | 90.95 | 43.03 | 58.4 | 38.07 | 23.83 |
| | | 50.64 | 20.27 | 67.64 | 90.56 | 41.18 | 56.46 | 37.05 | 26.07 |
| | | 50.72 | 20.48 | 67.10 | 90.65 | 39.11 | 55.69 | 36.00 | 24.31 |
| | | 50.49 | 20.09 | 67.15 | 90.77 | 38.78 | 55.81 | 37.38 | 25.27 |
| | Result | 50.74 ± 0.36 | 20.24 ± 0.15 | 67.38 ± 0.24 | 90.85 ± 0.31 | 41.13 ± 2.19 | 56.93 ± 1.33 | 36.92 ± 0.87 | 24.35 ± 1.46 |

A.8   EFFECTS OF $\mathcal{RN}$ AND $\mathcal{DA}$ ON COARSE-GRAINED AND FINE-GRAINED TASKS

From Table 2 and Table 3, we observe that $\mathcal{RN}$ module is more effective than $\mathcal{DA}$ module for coarse-grained models, but not as well as $\mathcal{DA}$ module for fine-grained models. There may be two reasons:

On the one hand, the default normalization of coarse-grained classification models is different from ImageNet and fine-grained classification models. Specifically, coarse-grained classification models use mean = [0.5,0.5,0.5] and std = [0.5,0.5,0.5], but ImageNet and and fine-grained classification models use mean = [0.485,0.456,0.406] and std = [0.229,0.224,0.225]; Therefore, using $\mathcal{RN}$ module can narrow normalization gap between ImageNet and coarse-grained domains.

On the other hand, the resolution of coarse-grained domains such as CIFAR-10 and CIFAR-100 is much lower than the ImageNet domain and fine-grained domains. Therefore, the intermediate features of images from coarse-grained domains are more coarse than those from ImageNet and fine-grained domains, which may enlarge the gap between ImageNet and coarse-grained domains.

