# OpenReview forum: "Beyond ImageNet Attack: Towards Crafting Adversarial Examples for Black-box Domains"
_ICLR.cc/2022/Conference — ICLR 2022 Poster_

### Official Review · Reviewer_9vb9 · 2021-10-19

**Correctness:** 3
**Technical Novelty And Significance:** 3
**Empirical Novelty And Significance:** 3
**Recommendation:** 6
**Confidence:** 4

**Details Of Ethics Concerns:**

This work can evaluate the vulnerability of deployed models. It does not need to know the details of deployed models and the information of training data to achieve the attack, which has certain hazards.

**Main Review:**

Strengths:
1. BIA focuses on disrupting low-level features to improve transferability.
2. This work proposes random normalization (RN) module to handle the various distributions between source domains and target domains.
3. This work proposes domain-agnostic attention (DA) module to produce a more robust feature representation.

Weaknesses:
1. RN module and DA module are not always mutually reinforcing. The reason behind this has not been analyzed.
2. In competitors, diverse inputs method (DIM) is not new. From this perspective, why not use the more powerful MI-DI-TI-FGSM or a newer transfer attack？
3. Table 2 and Table 3 show the transferability comparisons on classification tasks. However, the effects of DA and RN seem to depend on different models. To understand more deeply, it is necessary to analyze why different modules have different effects in different models.

Minor questions:
1. What are the experiments on the fine-grained and coarse-grained classification to prove? Why distinguish between fine-grained and coarse-grained? There is no clear explanation in this work.
2. Does the comparison method differ greatly in training costs?


**Summary Of The Paper:**

This paper focuses on the transferability of black-box domains. In real life, we do not know the relevant information of the deployed model and transfer attacks on black-box domains can better evaluate the vulnerability of deployed models. Therefore, Beyond ImageNet Attack (BIA) is proposed to investigate the transferability towards black-box domains (unknown classification tasks) with the only knowledge of the ImageNet domain. From the perspective of data and model, the authors propose random normalization (RN) module and domain-agnostic attention (DA) module to narrow the gap between the source and target domains. Finally, BIA achieves state-of-the-art performance in black-box domains settings.

**Summary Of The Review:**

I tend to accept this paper because it focuses on more realistic black box attack settings and proposes two modules to improve performance. The design of the module is insightful and effective, but the module proposed under some models is not always effective, which limits its application and requires more adequate analysis.

---

> ### Author Response · Authors · 2021-11-15
> **Response to Reviewer 9vb9**
>
> **Thanks for your encouraging remark!  Please see our generic comments and the detailed replies below:**
>
> _**Q1 "RN module and DA module are not always mutually reinforcing."**_
>
> **A1**: Please refer to the response to generic comment #2.
>
> _**Q2 "In competitors, diverse inputs method (DIM) is not new. From this perspective, why not use the more powerful MI-DI-TI-FGSM or a newer transfer attack？"**_
>
> **A2**: In our paper, DIM denotes a combination of DI-FGSM and MI-FGSM. We also tested the performance of TI-DIM (i.e. MI-DI-TI-FGSM) before. However, it is less effective than DIM (see below Table e). Therefore, in our paper, we instead report the result of DIM.
>
> Table e: Results for DIM and TI-DIM.
>
> | Attack|CIFAR-10| CIFAR-100| STL-10| SVHN| CUB-200-2011|Stanford Cars| FGVC Aircraft|
> |:------:|:------:|:------:|:------:|:--------:|:------:|:------:|:------:|
> | DIM| **77.16**| **44.75**| **72.74**|**91.53**| **62.86**| **78.10**| **66.88**|
> | TI-DIM| 88.84| 61.52| 75.24| 92.23| 66.24| 82.88| 73.96|
>
> _**Q3 "The effects of DA and RN seem to depend on different models. To understand more deeply, it is necessary to analyze why different modules have different effects in different models."**_
>
> **A3**: From Table 2 and Table 3, we observe that $\mathcal{RN}$ module is more effective than $\mathcal{DA}$ module for coarse-grained models, but not as well as $\mathcal{DA}$ module for fine-grained models. There may be two reasons:
> - The default normalization of coarse-grained classification models is different from ImageNet and fine-grained classification models. Specifically, coarse-grained classification models use mean = [0.5,0.5,0.5] and std = [0.5,0.5,0.5], but ImageNet and and fine-grained classification models use mean = [0.485,0.456,0.406] and std = [0.229,0.224,0.225]; Therefore, using $\mathcal{RN}$ module can narrow normalization gap between ImageNet and coarse-grained domains.
> - The resolution of coarse-grained domains such as CIFAR10 and CIFAR100 is much lower than the ImageNet domain and fine-grained domains. Therefore, the intermediate features of images from coarse-grained domains are more coarse than those from ImageNet and fine-grained domains, which may enlarge the gap between ImageNet and coarse-grained domains.
>
> _**Q4 "What are the experiments on the fine-grained and coarse-grained classification to prove? Why distinguish between fine-grained and coarse-grained? "**_
>
> **A4**: In general, the image classification tasks can be roughly divided into two categories in terms of label granularity [j]: fine-grained and coarse-grained. The fine-grained image classification task focuses on differentiating between hard-to-distinguish object classes, such as species of birds, flowers, or animals; and identifying the makes or models of vehicles. In contrast, coarse-grained classification is a fundamental task that attempts to comprehend an entire image as a whole. Take an image of a cat, for example, fine-grained tasks focus more on distinguishing which kind of cat it is while in coarse-grained, we just need to recognize it as a cat rather than a dog. In this paper, since we focus on the transferability of unseen classification tasks, we experiment on both to find out if the proposed method can generalize well on these two tasks.
>
> _**Q5 "Does the comparison method differ greatly in training costs?"**_
>
> **A5**: Most attacking methods can be divided into generative-models-based and iterative-optimization-based approaches [k]. For the generative-models-based methods such as CDA and our BIA, the generator needs to be trained first. When attacking, it only needs **one forward propagation** on the trained generator to get the adversarial examples. In contrast, iterative-optimization-based methods such as DIM and DR need  **tens or even hundreds of forward&backward propagation** on a substitute model to generate adversarial examples. Thus, our BIA requires the same training costs with CDA since the costs of $\mathcal{RN}$ and $\mathcal{DA}$ are negligible, but it is much faster when attacking than DIM and DR.
>
> References:
>
> [j] Touvron et al. Grafit: Learning fine-grained image representations with coarse labels. (ICCV 2021)
>
> [k] Naseer et al. Cross-domain transferability of adversarial perturbations. (NeurIPS 2019)

---

> > ### Comment · Reviewer_9vb9 · 2021-11-22
> > **Thanks for the response**
> >
> > Q1: Thank you for the response and address this concern. However, I think the results with combining the two proposed techniques (RN and DA) should be present in the main draft instead of the appendix.
> >
> > Q2: The novel transfer attack should also be tried, such as NI-SI-FGSM and VMI-FGSM.
> >
> > Q3&Q4: I hope this part of the analysis will appear in the final version.
> >
> > Q5: The response seems to pay more attention to inference cost rather than training cost.
> >
> > Summary:
> > The response addresses part of my concerns, but not all of them, so I keep the score. These analyses are very important for understanding this work, so I hope the authors consider the suggestions and integrate them into the final version.

---

> > > ### Author Response · Authors · 2021-11-22
> > > **Thanks for your response**
> > >
> > > Thank you a lot for your response. According to your suggestion, we have made some improvements (highlighted in magenta color) in the latest revision ad follows:
> > > - move the combining result of RN and DA to the main draft, and at the same time, add the corresponding discussion in generic comments #2 to Section 4.2.
> > > - add the analysis in Q3&Q4.
> > >
> > > _**Response to novel transfer attacks and training cost:**_
> > >
> > > Transfer attacks: The experiments of NI-SI-FGSM and VMI-FGSM have been launched. But due to the expensive time cost for such iteration-based methods, we need more time to evaluate them. Specifically, compared to iterative attacks introduced in our manuscript (e.g., PGD and DIM), NI-SI-FGSM and VMI-FGSM require approximately 4$\times$ and 19$\times$ more computational overhead, respectively. However, even using PGD would take more than 40 hours (on one V100 GPU) to generate adversarial examples for all domains on a substitute model. We will try our best to accelerate this evaluation.
> > >
> > > Training cost: the generator is trained on ImageNet (1.2millon images) for one epoch,  requiring about 10~12 hours on one V100 GPU.

---

> > > > ### Author Response · Authors · 2021-11-26
> > > > **Experiments for novel transfer attacks**
> > > >
> > > > _**Experiments for novel transfer attacks**_
> > > >
> > > > To accelerate the evaluation for novel transfer attacks in our practical black-box scenario, we randomly sampled 1,000 images of each black-box dataset and perform attacks on them. As demonstrated in Table h, the state-of-the-art transfer attacks such as NI-SI-FGSM [l] and VMI-FGSM [m] cannot get good results in our practical black-box scenario. For example, the average top-1 accuracy of CUB-200-2011 models still maintains 77.0% and 70.1% (average of models with different backbones) after SI-NI-FGSM and VMI-FGSM attacks respectively, while our BIA+$\mathcal{DA}$ can effectively drop the top-1 accuracy to 39.4%.
> > > >
> > > >
> > > > Table h. Results for crafting adversarial examples via VGG16 (the lower, the better).
> > > >
> > > > | Attacks 	| Cifar10 	| Cifar100 	| STL10 	| SVHN 	| CUB-avg 	| Cars-avg 	| Aircraft-avg 	| AVG. 	|
> > > > |:---:	|:---:	|:---:	|:---:	|:---:	|:---:	|:---:	|:---:	|---	|
> > > > |Clean | 94.5|	73.6|	76.6	|94.8| 87.7	|93.5|	91.7| 87.5|
> > > > | SI-NI-FGSM 	| 75.8 	| 44.8 	| 71.3 	| 88.7 	| 77.0 	| 87.7 	| 79.9 	| 75.0 	|
> > > > | VMI-FGSM 	| 75.3 	| 46.1 	| 72.7 	| 91.0 	| 70.1 	| 78.6 	| 69.2 	| 71.9 	|
> > > > | BIA 	| 57.6	| 22.7 	| 68.8 	| 88.7 	| 46.4 	| 59.8 	| 45.3 	| 55.6 	|
> > > > | BIA+$\mathcal{DA}$ 	| 55.3 	| 21.8 	| 68.7 	| 90.7 	| **39.4** 	| **48.1** 	| **35.0** 	| **51.3** 	|
> > > > | BIA+$\mathcal{RN}$	| **54.3** 	| **20.9** 	| **67.3** 	| **87.6** 	| 42.9 	| 49.1 	| 38.5 	| 51.5 	|
> > > >
> > > > ### Reference
> > > >
> > > > [l] Lin et al. Nesterov accelerated gradient and scale invariance for improving transferability of adversarial examples (ICLR 2020)
> > > >
> > > > [m] Wang et al. Enhancing the Transferability of Adversarial Attacks through Variance Tuning (CVPR 2021)

---

### Official Review · Reviewer_bvNx · 2021-10-30

**Correctness:** 3
**Technical Novelty And Significance:** 3
**Empirical Novelty And Significance:** 3
**Recommendation:** 6
**Confidence:** 4

**Main Review:**

See the pros/cons below.

### Pros
1. Considering more practical threat model is certainly helpful and important for the transfer attack research.
2. The results indeed demonstrate large improvement of BIA in terms of error rate.

### Cons
1. I'm a little concerned about the "cross-domain" statement made in this work. To me, the target datasets considered in this work (CIFAR, STL, CUB, Stanford Cars) are still coming from the same natural imagery "domain" as ImageNet, despite they have different label spaces. In particular, CUB is known to have overlap with ImageNet [1], where the "cross-domain" claim certainly does not hold. An example case that is more "cross-domain" would be to transfer from ImageNet model to a ChestX-ray model (in a similar sense to Naseer et al.).

2. The specific methodology of BIA seems not new. It is known that perturbing intermediate layer features can yield more transferable adversarial examples (e.g., [2]). In fact, the formulation of BIA appears very similar to the one proposed in [2] (while BIA minimizes the cosine similarity between intermediated layer features of the clean and adversarial examples, [2] maximizes the euclidean distance, which essentially is the same). Feature space attacks are also shown to be more powerful than decision space attacks in more strict black-box transfer scenarios [3], but Sec. 3.2 fails to recognize these existing works. Clearly identifying the difference between BIA and [2] might help address this concern.

3. If my above judgement of BIA not being new is correct, then my further concern comes from the result side. In Table 2, it seems that the performance gain can be largely attributed to the BIA (or essentially feature space attack) itself rather than the DA and RN module. This hurts the empirical novelty to some extent, as previous works have shown the superiority of feature space attacks in either standard or more strict transfer settings ([2,3]).


[1] http://www.vision.caltech.edu/visipedia/CUB-200-2011.html

[2] Feature Space Perturbations Yield More Transferable Adversarial Examples

[3] Perturbing Across the Feature Hierarchy to Improve Standard and Strict Blackbox Attack Transferability

**Summary Of The Paper:**

This work first identifies a more practical threat model for black-box transfer adversarial attack, where the target model's domain remains unknown, and the attacker's surrogate model may be trained in another domain. Then, the BIA attack is proposed to enhance transferability, whose key idea is to distort low-level features captured by DNN's intermediate layers instead of perturbing the domain-specific features in the output layer. Two modules, DA and RN, are further proposed to improve attack success rate. Experimental results demonstrate that BIA is more effective than existing methods.

**Summary Of The Review:**

This paper indeed identifies a more practical threat model, but the experiments do not closely match the proposed "cross-domain" scenario, and the performance gain seems to largely come from existing technique (perturbing feature space instead of decision space). These issues prevent me from recommending for acceptance.

---

> ### Author Response · Authors · 2021-11-15
> **Response to Reviewer bvNx**
>
> **Thank you for carefully reviewing our paper and raising these important questions. Please see our generic comments and the detailed replies below:**
>
> _**Q1 The "Cross-domain" statement.**_
>
> **A1**: Actually, we think that "cross-domain" is appropriate in this case. Take person re-identification for an example, researchers usually take the change from one scenario to another one as "cross-domain" even if all the focused objects of different cameras are persons [h][i].
>
> _**Q2 "Difference between BIA and [2]"**_
>
> **A2**: Please refer to the response to generic comment #1. More detailed differences with [2] are listed as follows.
> - Motivation difference: [2] requires the dataset knowledge of the target domain. In contrast, our method only requires the knowledge from source domains to make the attack more practical.
> - Method difference: [2] is a transfer-based attack that optimizes adversarial examples by attacking the feature layer. Our method also adopts this strategy as our baseline to ensure the good performance from the beginning. However, this strategy is not always effective and we propose two modules  $\mathcal{RN}$ and $\mathcal{DA}$ to improve the attack transferability further.
> - Efficiency difference: [2] crafts adversarial examples via iterative optimization on a deep substitute model, and the computation cost is expensive at the attacking phase. In contrast, our BIA leverages a generator to learn an adversarial function. After training, adversarial examples can be efficiently generated with only one inference on a relatively shallow generator.
>
> _**Q3 "It seems that the performance gain can be largely attributed to the BIA (or essentially feature space attack) itself rather than the DA and RN module."**_
>
> **A3**: Please refer to the response to generic comment #1.
>
> References:
>
> [h] Lin et al. Cross-domain visual matching via generalized similarity measure and feature learning. (TPAMI 2016)
>
> [i] Liu et al. Adaptive transfer network for cross-domain person re-identification. (CVPR 2019)

---

> > ### Comment · Reviewer_bvNx · 2021-11-20
> > **Thanks for the response**
> >
> > ***Q1 The "Cross-domain" statement.***
> >
> > I have to say I'm not convinced by this example, especially given that it is from another research topic which does not appear to have any strong relationship with the black-box transfer attack research.
> >
> > The reason that I think one should be careful about the "cross-domain" statement is because essentially we are seeking for powerful attacks when reducing the attacker's knowledge about the target dataset to the lowest level (which I believe is also the core motivation of this paper). However, one could argue that this is not the case in the scenarios studied by the paper, i.e., the attacker still has certain amount of (non-trivial) knowledge about the target dataset. For example, ImageNet certainly covers the concept of all CIFAR-10 categories despite their resolution difference. Also, the authors of CUB explicitly warn that CUB images overlap with ImageNet (as I mentioned in the last comment). The consequence of drawing conclusions from these less "cross-domain" situations is that, the model owners may be overly optimistic and think that their models are at risk only if the attacker has access to a similar dataset. This is why I think ideally a more "cross-domain" experiment such as ImageNet -> ChestXray should be presented (actually is it possible for the authors to do this experiment?); or at least one should be careful about what current scenarios fail to approximate.
> >
> > ***Q2 "Difference between BIA and [2]"***
> >
> > I appreciate the authors' clarification on this, but essentially it seems that my previous judgement is correct, that methodology-wise the base component of BIA (without the DA and RN module) is not new. With that being said, I do like the paragraph presented in the generic comment #1 (particularly this sentence "As a baseline for this new problem, BIA contains two components that can help to generate transferable adversarial examples according to existing literature: the generative model [a] and the intermediate feature disruption [b][c][d].") This identifies the contributions/novelties of this work much more clearly than what's currently being presented in the manuscript, and I would like to see it being included in Section 3.2 when discussing Equation (2).
> >
> > ***Q3 "It seems that the performance gain can be largely attributed to the BIA (or essentially feature space attack) itself rather than the DA and RN module."***
> >
> > The Table a presented in generic response #1 (especially the AVG. column) in part helps address the concern. Consider put this type of AVG statistics in those existing tables in the paper.
> >
> > ***Summary***
> >
> > I'm raising my score to 6, given that the new components (DA and RN) indeed lead to further gains compared to existing techniques and the considered threat model is of importance. Yet, I strongly recommend the authors to consider the suggestions in my comments and integrate them in the paper.

---

> > > ### Author Response · Authors · 2021-11-22
> > > **Thanks for your response**
> > >
> > > Thank you a lot for your response. According to your suggestion, we have made some improvements (highlighted in magenta color) in the latest revision ad follows:
> > > - put the AVG statistics in those existing tables in the paper.
> > > - include the description of the main contributions/novelties of this work such as "As a baseline for this new problem, BIA contains two components that can help to generate transferable adversarial examples according to existing literature: the generative model [a] and the intermediate feature disruption [b][c][d]" in Section 3.2 when discussing Equation (2). On top of this, we also revised the introduction to make our contributions more clear.
> > >
> > > _**Response to the "cross-domain" statement:**_
> > >
> > > Thank you for your suggestion. The result for ImageNet -> ChestXray experiment is shown in Table f, where our substitute model is Res-152 and the target model is the same as CDA (i.e. a Dense-121 network trained to diagnose pneumonia). It can be observed that our method still shows good transferability in this case, validating its effectiveness on cross-domain scenarios.
> > >
> > > Besides, we also conducted another experiment to evaluate the "cross-domain" scenario (CUB-200-2011 -> others). As shown in answer A4 to reviewer YKsk (#1) (can also be found in Table 5 in the revision),  our adversarial example generators trained on CUB-200-2011 (bird species classification) can still show good attack transferability to models for other classification tasks such as cars classification (Standford Cars Dataset), aircarft classification (FGVC Aircraft Dataset), etc.  For example, the top-1 accuracy of the Standford cars model drops to 54.65%. Compared to existing SOTA methods CDA, excluding the gains from feature attack, our proposed DA module helps to improve the attack success rate by about 13%. This shows the attack effectiveness of the proposed method.
> > >
> > > Table f. The top-1 accuracy (%) for ImageNet->ChestXray after attacking. The generators were trained against ImageNet.
> > >
> > > |   Attack   | ImageNet->ChestXY     |
> > > |:---:    |:---:    |
> > > | clean     | 77.45     |
> > > | CDA     | 57.71     |
> > > | BIA     | 57.10     |
> > > | BIA+DA     | 55.98     |
> > > | BIA+RN     | 47.30     |

---

### Official Review · Reviewer_YKsk · 2021-11-03

**Correctness:** 3
**Technical Novelty And Significance:** 2
**Empirical Novelty And Significance:** 3
**Recommendation:** 8
**Confidence:** 4

**Main Review:**

### Strengths:

1) The problem setting (no access to target data) is of importance - in practice, access to data is as hard, if not harder, than access to model.
2) The experiments are extensive, and clearly show a significant improvement in black-box attack capability.
3) The code provided with the paper, along with the appendix, help gaining a clearer understanding of the method (conversely, they further emphasize readability issues of the manuscript).

### Weaknesses:

1) The manuscript is poorly written - grammatical mistakes and semantic mistakes are aplenty. Some phrases are the opposite of what the method actually does. Section 3.4 states "Specifically, we apply a channel-wise average pooling to the feature maps at layer L" where as the actually operation is cross-channel average pooling (Refer to [1]). Other mistakes are highlighted below.

2) The novelty of the method is limited. In Wen Zhou et al., 2018 [2], Intermediate feature disruption is used to increasing black-box transferability. In Weibin Wu et al., 2020 [3], attention is used for increasing transferability. The paper does not mention these works.

3) The claims of section 4.2 are only weakly supported. Statement: "The downsampling module has an essential impact on the resulting adversarial examples" I fail to see how this can be inferred from visualizing the cross-channel attention outputs.

### Other weaknesses:

1) Since all the experiments are only regarding ImageNet -> target datasets, it is unclear how well the method will perform if the source dataset is different (especially if the source dataset is small).
2) The metrics do not include standard deviation across multiple random runs. Evaluating the standard deviation in at least one setting will elucidate the significance of the results.
3) Results with combining the two proposed techniques (RN and DA) *should* be present in the main draft. This is an important question that the manuscript only deals with in the appendix. The manuscript with benefit from a discussion on the fact that using these two techniques in tandem is challenging, and fails to consistently out-perform using just a single module.

### Text Errors:

#### Abstract:
1) transferability nature ==> transferable nature.
2) the only knowledge ==> only the knowledge
3) the coarse-grained domain ==> coarse-grained domains

#### Introduction:

1) possible to the spotlight ==> possible
2) transparent ==> opaque
3) the query ==> querying
4) but more threatening ==> and more threatening
5) the generator ==> a generator

#### Method:
1) they can subject to the ==> they can be modeled as samples from the standard normal distribution.
2) even the inputs are not ==> even if the inputs are not

#### Experiments:
1) in the Torchvision. ==> in the Torchvision library.
2) another seven ==> seven other

### Questions for the authors:

1) How will the generator network perform with its trained with all source models at once? (See experiments - Table 3 in Konda Reddy Mopuri et al., 2017 [4]) I suspect that it should further increase the transferability.

2) Have the authors tried to increase the RGB jittering when comparing to existing methods? I suspect that with significant jittering, augmentation may perform similar to random normalization.

### References:

[1] Network In Network, Min Lin, Qiang Chen, Shuicheng Yan, arXiv 2013.

[2] Transferable Adversarial Perturbations, Wen Zhou, Xin Hou, Yongjun Chen, Mengyun Tang, Xiangqi Huang, Xiang Gan, Yong Yang, ECCV 2018.

[3] Boosting the Transferability of Adversarial Samples via Attention, Weibin Wu, Yuxin Su, Xixian Chen, Shenglin Zhao, Irwin King, Michael R. Lyu, Yu-Wing Tai, CVPR 2020.

[4] NAG: Network for Adversary Generation, Konda Reddy Mopuri, Utkarsh Ojha, Utsav Garg, R. Venkatesh Babu, CVPR 2018

**Summary Of The Paper:**

This paper tackles the challenge of generating adversarial perturbation for a target model - with no access to the model, or the model's training data (i.e. target domain). Using a trained model and data from a source domain (ImageNet), the authors train a generator to craft perturbations which maximize the cosine distance between the intermediate features of clean and adversarial images. This generator is then assisted by two techniques - random normalization of the input image, and spatial attention on Intermediate-layer features (used for cosine distance). Experiments show that this method outperforms prior methods in black-box setting (no access to target domain or model) as well as white-box setting.

**Summary Of The Review:**

The method proposed in this paper out-performs existing methods, and targets an important setting (no access to target domain or model). However, the writing is error-ridden, and the proposed method is only marginally novel w.r.t. existing works. Therefore, I rate the paper as marginally above accept threshold, conditional on the authors correcting the mistakes highlighted above.

---

> ### Author Response · Authors · 2021-11-15
> **Response to Reviewer YKsk (1 of 2)**
>
> **Thank you for your positive feedback and insightful comments. Please see our generic comments and the detailed replies below:**
>
> _**Q1  "Writing"**_
>
> **A1**: Thank you for raising these writing problems. We have revised them and checked the grammatical and semantic errors of the entire paper carefully in the revision to improve its readability.
>
> _**Q2 "Novelty"**_
>
> **A2**:  Please refer to the response to generic comment #1.
>
> _**Q3 "The claims of section 4.2 are only weakly supported. Statement: "The downsampling module has an essential impact on the resulting adversarial examples" I fail to see how this can be inferred from visualizing the cross-channel attention outputs".**_
>
> **A3**: Figure 5 visualizes the outputs of each block in the trained generative models. We can find that the downsampling module tries to find out the discriminative parts of the input image for adding adversarial perturbations later. This can help to improve the transferability for cross-domain attacks. We have included more discussion in the revision.
>
> _**Q4 "Since all the experiments are only regarding ImageNet -> target datasets, it is unclear how well the method will perform if the source dataset is different (especially if the source dataset is small)".**_
>
> **A4**: In the following Table c, we report the results for CUB-200-2011->Others. Generators are learned against CUB-200-2011 domain (the substitute model is DCL (backbone: Res-50) and training data is the test data from CUB-200-2011 (5,794 images)). For results of fine-grained domains, we average top-1 accuracy of target models with backbone SENet-154 and SE-Res101. For the result of the ImageNet domain, we average top-1 accuracy of all models introduced in our manuscript. It can be observed that our method consistently outperforms our main competitor CDA by a large margin. Besides, we also notice that transferring from CUB-200-2011->CIFAR is very challenging (compared with our reported results in Table 2). Therefore, we highlight the necessity of using a large-scale dataset such as ImageNet to train the adversarial examples generator.
>
> Table c. The results for CUB-200-2011->Others.
>
> |Attacks|Cifar10|Cifar100|STL10|SVHN|CUB-200-2011|Stanford Cars|FGVC Aircraft|ImageNet|
> |:---:|:---:|:---:|:---:|:---:|:---:|:---:|:---:|:---:|
> |CDA|83.61|54.83|70.49|91.81|64.34|74.37|71.17|41.05|
> |BIA|**82.62**|54.57|71.43|91.67|40.40|68.25|57.90|33.83|
> |BIA+$\mathcal{DA}$|82.94|54.15|71.70|**87.28**|**29.55**|**54.65**|55.15|**29.82**|
> |BIA+$\mathcal{RN}$|83.84|**53.63**|**69.79**|87.95|42.88|57.01|**50.85**|30.09|

---

> > ### Author Response · Authors · 2021-11-15
> > **Response to Reviewer YKsk (2 of 2)**
> >
> > _**Q5 "Standard deviation across multiple random runs".**_
> >
> > **A5**: Thanks for your insightful suggestion. In Table a of the generic comment, we take BIA (against VGG-16) as an example to show the results of multiple random runs. It can be observed that the standard deviation across multiple random runs is not significant. We have also included the standard deviation in our revision.
> >
> > _**Q6 "Results with combining the two proposed techniques (RN and DA)".**_
> >
> > **A6**: Please refer to the response to generic comment #2.
> >
> > _**Q7 "How will the generator network perform with its trained with all source models at once?"**_
> >
> > **A7**: The following Table d shows the results for training against Vgg-16 and an ensemble of Vgg-16, Vgg-19, Res-152 and Dense-169, respectively. It can be observed that ensemble-based training can significantly improve the transferability of adversarial examples towards fine-grained domains, which is consistent with conclusions in previous studies[e][f].
> >
> > Table d:  Results for training against Vgg-16 and an ensemble of Vgg-16, Vgg-19, Res-152 and Dense-169.
> >
> > |Method|CIFAR10|CIFAR100|STL10|SVHN|CUB|CARS|Aircraft|
> > |:---:|:---:|:---:|:---:|:---:|---|---|---|
> > |BIA|57.38|22.47|69.45|90.44|52.99|69.90|60.31|
> > |BIA (ensemble)|**54.96**|**21.73**|**68.94**|**85.85**|**29.15**|**46.47**|**36.87**|
> > |BIA+DA|55.16|21.71|70.00|91.76|40.27|59.48|47.91|
> > |BIA+DA (ensemble)|**52.97**|**20.51**|**69.51**|**89.15**|**18.47**|**38.99**|**25.41**|
> > |BIA+RN|52.81|**21.12**|67.55|88.03|46.15|62.64|52.54|
> > |BIA+RN (ensemble)|**50.99**|21.95|**66.06**|**81.66**|**35.40**|**41.87**|**27.81**|
> >
> > _**Q8 "Have the authors tried to increase the RGB jittering when comparing to existing methods? I suspect that with significant jittering, augmentation may perform similar to random normalization."**_
> >
> > **A8**: In fact, we think that the proposed $\mathcal{RN}$ module can be regarded as a kind of RGB jittering. In contrast to the traditional color jittering (i.e., randomly changing the brightness, contrast, hue and saturation) [g]  which is mainly performed in HSV space, our $\mathcal{RN}$ performs random normalization in RGB space. Actually, we have conducted experiments to find out if the RN module is equivalent to color jittering. The result in Figure 9 of the appendix shows that augmenting with color jittering (AUG) is less effective than $\mathcal{RN}$ module, and sometimes AUG even degrades performance (e.g. second subfigure in Figure 9).  We think that the "color jittering" will change the texture of input images and lead the attack model to capture the shape cues to fool the substitute model.  However, the $\mathcal{RN}$ module only destroys the color information, which leads the attack model to learn texture and shape cues from the model.
> >
> > References:
> >
> > [e] Liu et al. Delving into transferable adversarial examples and black-box attacks. (ICLR 2017)
> >
> > [f] Mopuri et al. NAG: Network for adversary generation. (CVPR 2018)
> >
> > [g] https://pytorch.org/vision/main/generated/torchvision.transforms.ColorJitter.html

---

> > > ### Comment · Reviewer_YKsk · 2021-11-20
> > > **Thank you for the detailed response!**
> > >
> > > Thank you for the detailed reply!
> > >
> > > **Q1** Writing:
> > >
> > > Thank you for correcting the manuscript.
> > >
> > > **Q2** Novelty:
> > >
> > > The primary contributions are indeed the RN and DA modules. These still have the following drawbacks:
> > > 1) The benefits from the modules are limited, and highly variable.
> > > 2) RN model is closely related to RGB jittering with limited novelty.
> > > 3) Using them together typically does not result in stronger attacks.
> > >
> > > **Q3**  section 4.2:
> > >
> > > Thank you for improving this section. Now, the draft presents this idea clearly.
> > >
> > > **Q4** CUB 200, 2011 -> Others
> > >
> > > Thank you for adding this result to Appendix A.6. One surprising result is CUB 200, 2011 -> CUB 200, 2011. The model is trained with the test images, with ResNet50 backbone, and performs black box attack on SENet-154 and SE-Res101 on the train images of CUB 200, 2011. I am surprised that the attack is fairly weak despite the data being from the same distribution.
> > >
> > > **Q5**: Standard deviation
> > >
> > > As the primary contributions of this paper are the RN and DA modules, what we are really interested in is the significance of these modules. While the standard deviation of the BIA model is handy, what really matter is the standard deviation of (BIA + RN)/(BIA + DA) model. This would allow us to utilize statistical tests such as Welch's t-test to evaluate the significance of the result.
> > >
> > > **O6**: Combining the two proposed techniques
> > >
> > > I note that the authors have discussed this in the appendix, and the main draft directs the reader to the corresponding appendices. I still believe "Results with combining the two proposed techniques (RN and DA) should be present in the main draft" (as mentioned in the review).
> > >
> > > **Q7**: Ensemble Model.
> > > Thank you for conducting this experiment.
> > >
> > > **Q8**: RN and RGB jittering.
> > > Thank you for your response.
> > >
> > > **Overall Comment**:
> > >
> > > Apart from my concerns regarding the novelty of the work, the authors have addressed all my concerns. In light of the improvements in the draft, I will raise my rating to "8: Accept, good paper".

---

> > > > ### Author Response · Authors · 2021-11-22
> > > > **Thanks for your response!**
> > > >
> > > > Thank you a lot for your response. According to your suggestions, we have made some improvements (highlighted in magenta color) in the latest revision as follows:
> > > > - add the standard deviation for BIA, BIA+RN, BIA+DA, BIA+RN+DA in Table 7.
> > > > - move the combining result of RN and DA to the main draft, and at the same time, add the corresponding discussion in generic comments #2 to Section 4.2.

---

### Author Response · Authors · 2021-11-15
**Revision and generic comments (1 of 3)**

We thank all the reviewers for their constructive comments and insightful suggestions. We carefully revised the manuscript according to the comments of all the reviewers. For convenience, we highlighted the revised text in color except for the revision of grammars. Here we briefly summarize the updates we have made to the revision:
- check and revise the grammatical and semantic errors of the entire paper carefully in the revision to improve its readability.
- cite and discuss the papers the reviewers provided.
- re-describe Figure 5 to make it more understandable.
- add discussions about the combination of $\mathcal{RN}$ and $\mathcal{DA}$ modules in Appendix A.5.
- add experiments for changing source domain from ImageNet to CUB-200-2011 to show the cross-domain transferability in Appendix A.6.
- add experiments for ensemble-model attacks in Appendix A.7.
- add experiments for standard deviation across multiple random runs in Appendix A.8

---

> ### Author Response · Authors · 2021-11-15
> **Revision and generic comments (2 of 3)**
>
> We have concluded all the comments from all the reviewers and responded to the generic comments as follows:
>
> _**#1 The novelty against intermediate feature disruption methods.**_
>
> **Response**: In this paper, we come up with a more practical black-box attack: whether there exists an attack method that can generate adversarial examples with good transferability on black-box domains even if we have only the knowledge of the source domain? For this, we proposed the Beyond ImageNet Attack (BIA) framework. As a baseline for this new problem, BIA contains two components that can help to generate transferable adversarial examples according to existing literature: the generative model [a] and the intermediate feature disruption [b][c][d]. This ensures a good attack performance for BIA from the beginning.
> However, BIA also fails to generalize well in some cases (Table a, highlighted in italic). For this, we further propose two variants to narrow the gap between the source and target domains from the data and model respectively, namely $\mathcal{RN}$ and $\mathcal{DA}$. These two components can help to improve the transferability towards black-box domains. As shown below in Table a, though the intermediate feature disruption technique helps to improve the transferability in most cases, our proposed $\mathcal{RN}$ and $\mathcal{DA}$ can also help to improve the performance a lot (about 5% on Vgg-16, 7% on VGG-19, 5% on ResNet152, 3% on DenseNet169).
> Meanwhile, to ensure the stability and credibility of the evaluations, experiments are repeated several times and the results are shown in Table b (only BIA to meet tight deadlines).
> We also provide some visualizations in Figure 6 from the interpretable perspective to make the results more explainable.
> Given all that, we would suggest that this work has its unique novelties and practical meaning in both conception and technique.
>
> Table a: The top-1 accuracy collected from Table 2 and Table 3. CUB-avg stands for the average top-1 accuracy of target models with different backbones (i.e. Res-50,  SENet-154 and SE-Res101).
>
> | Model| Attacks| **Cifar10**| **Cifar100**| **STL10**| **SVHN**| **CUB-avg**| **Cars-avg**| **Aircraft-avg**| **AVG**.|
> | ---| ---| ---| ---| ---| ---| ---| ---| ---| ---|
> | VGG-16| CDA| 66.41| 32.37| 72.91| 92.17| 67.73| 77.68| 64.42| 67.67|
> || BIA (Ours)| 57.38| 22.47| 69.45| 90.44| 47.92| 59.89| 45.38| 56.13|
> || BIA+DA (Ours)| 55.16| 21.71| 70.00| 91.76| **39.50**| **47.75**| **34.70**| 51.51|
> || BIA+RN (Ours)| **52.81**| **20.82**| **67.55**| **88.03**| 42.45| 48.88| 38.17| **51.24**|
> | VGG-19| CDA| 81.60| 51.53| 71.43| 92.64| 63.02| **_69.98_**| 57.92| 69.73|
> || BIA (Ours)| 57.88| 23.12| 69.84| 88.89| 52.57| 71.29| 52.26| 59.41|
> || BIA+DA (Ours)| 57.26| 23.04| 70.16| 90.08| **38.40**| **53.69**| **35.93**| **52.65**|
> || BIA+RN (Ours)| **54.47**| **22.61**| **68.23**| **88.08**| 42.09| 59.33| 41.93| 53.82|
> | Res-152| CDA| 66.47| 39.30| **_69.81_**| **_88.09_**| 50.57| 64.62| 56.91| 62.25|
> || BIA (Ours)| 65.49| 33.48| 69.91| 89.46| 49.53| 50.99| 40.71| 57.08|
> || BIA+DA (Ours)| 65.34| **32.68**| 69.65| 91.38| 39.45| 51.29| **34.07**| 54.84|
> || BIA+RN(Ours)| **61.23**| 32.84| **68.04**| **85.79**| **38.34**| **40.30**| 37.36| **51.99**|
> | Dense-169| CDA| **_67.75_**| **_35.03_**| **_69.00_**| 88.76| 56.97| 67.60| 61.16| 63.75|
> || BIA (Ours)| 72.02| 38.99| 69.80| 86.12| 30.07| 34.37| 23.25| 50.66|
> || BIA+DA (Ours)| 71.69| 38.95| 70.60| 88.02| **25.82**| **25.74**| **10.68**| **47.36**|
> || BIA+RN (Ours)| **66.67**| **34.41**| **68.79**| **81.54**| 29.93| 31.55| 21.67| 47.79|
>
> Table b:  Standard deviation across multiple random runs. (_**11.18 Update: results for $\mathcal{RN}$ and $\mathcal{DA}$ variants have been added in Table 7 in our latest revision**_)
>
> | BIA|Cifar10| Cifar100| STL10| SVHN| CUB| CARS| Aircraft| Imagenet|
> |:---:|:---:|:---:|:---:|:---:|:---:|:---:|:---:|:---:|
> | Paper report| 57.38| 22.47| 69.45| 90.44| 52.99| 69.90| 60.31| 42.98|
> | random runs-1| 56.75| 21.86| 69.61| 90.48| 52.47| 68.13| 61.45| 44.04|
> |random runs-2| 57.31| 22.32| 69.83| 90.35| 52.04| 69.48| 57.37| 41.67|
> |random runs-3| 57.13| 22.82| 70.06| 90.01| 53.27| 69.74| 58.54| 41.02|
> |random runs-4| 57.03| 22.38| 70.05| 90.43| 53.04| 71.91| 62.02| 43.09|
> |random runs-5| 57.36| 22.50| 69.55| 89.78| 51.48| 68.15| 58.36| 42.00|
> | Result | 57.16$\pm$0.24| 22.39$\pm$0.31| 69.76$\pm$0.26| 90.25$\pm$0.29| 52.55$\pm$0.69| 69.55$\pm$1.39| 59.68$\pm$1.86| 42.47$\pm$1.10|
>
>
> References:
>
> [a] Baluja S, Fischer I. Adversarial transformation networks: Learning to generate adversarial examples. arXiv preprint arXiv:1703.09387, 2017.
>
> [b] Yosinski et al. How transferable are features in deep neural networks? (NeurIPS 2014)
>
> [c] Zhou et al. Transferable adversarial perturbations. (ECCV 2018)
>
> [d] Inkawhich et al. Feature space perturbations yield more transferable adversarial examples (CVPR 2019).

---

> > ### Author Response · Authors · 2021-11-15
> > **Revision and generic comments (3 of 3)**
> >
> > _**#2. The combination of RN and DA modules.**_
> >
> > **Response**: As shown in Figure 10 in the appendix, the combination of $\mathcal{RN}$ and $\mathcal{DA}$ modules out-performs using just one single module in most cases. However, it fails to consistently out-perform in all the cases. Specifically, when training against Dense-169, the transferability of generated adversarial examples degrades in fine-grained classification tasks. For this, we visualize the cross-channel average pooling of intermediate features for Dense-169 to better explain this phenomenon. As shown in Figure 11 of the revision, the $\mathcal{RN}$ module reinforces the discriminative features in Vgg16. However, it inhibits the response of objects' essential features extracted by Dense-169. Consequently, $\mathcal{DA}$ module may cause the resulting generator to reduce the ability to attack essential features, thereby making it challenging to use these two techniques in tandem.

---

### Decision · Program_Chairs · 2022-01-20

**Decision:**

Accept (Poster)

**Comment:**

This paper considers that the model's training data may be not accessible when learning the attacking model, and thus a more practical blackbox attack scheme, Beyond ImageNet Attack (BIA) framework, is designed. All the reviewers agreed that the setting in this paper is important and helpful when designing attack methods. However, the method is not totally new. Nevertheless, considering the importance of the problem investigated in this paper, the nice design of the overall framework, and the extensive experiments, the AC recommends accept for this paper.